# OFF-POLICY PRIMAL-DUAL SAFE REINFORCEMENT LEARNING

**Zifan Wu**[1], **Bo Tang**[23]\*, **Qian Lin**[1]\*, **Chao Yu**[1]†,
**Shangqin Mao**[3], **Qianlong Xie**[3], **Xingxing Wang**[3], **Dong Wang**[3]
[1]Sun Yat-sen University, Guangzhou, China
{wuzf5,linq67}@mail2.sysu.edu.cn, yuchao3@mail.sysu.edu.cn
[2]Institute for Advanced Algorithms Research, Shanghai, China
tangb@iaar.ac.cn
[3]Meituan, Beijing, China
{maoshangqin,xieqianlong,wangxingxing04,wangdong07}@meituan.com

## ABSTRACT

Primal-dual safe RL methods commonly perform iterations between the primal update of the policy and the dual update of the Lagrange Multiplier. Such a training paradigm is highly susceptible to the error in cumulative cost estimation since this estimation serves as the key bond connecting the primal and dual update processes. We show that this problem causes significant underestimation of cost when using off-policy methods, leading to the failure to satisfy the safety constraint. To address this issue, we propose *conservative policy optimization*, which learns a policy in a constraint-satisfying area by considering the uncertainty in cost estimation. This improves constraint satisfaction but also potentially hinders reward maximization. We then introduce *local policy convexification* to help eliminate such suboptimality by gradually reducing the estimation uncertainty. We provide theoretical interpretations of the joint coupling effect of these two ingredients and further verify them by extensive experiments. Results on benchmark tasks show that our method not only achieves an asymptotic performance comparable to state-of-the-art on-policy methods while using much fewer samples, but also significantly reduces constraint violation during training. Our code is available at https://github.com/ZifanWu/CAL.

## 1 INTRODUCTION

Reinforcement learning (RL) has achieved tremendous successes in various domains (Silver et al., 2016; Andrychowicz et al., 2020; Jumper et al., 2021; Afsar et al., 2022). However, in many real-world tasks, interacting with the environment is expensive or even dangerous (Amodei et al., 2016). By encoding safety concerns into RL via the form of constraints, safe RL (Gu et al., 2022) studies the problem of optimizing the traditional reward objective while satisfying certain constraints. In recent years, safe RL has received increasing attention due to the growing demand for the real-world deployment of RL methods (Xu et al., 2022b).

As one of the most important safe RL methods, primal-dual methods convert the original constrained optimization problem to its dual form by introducing a set of Lagrange multipliers (Chow et al., 2017; Tessler et al., 2018; Ray et al., 2019; Ding et al., 2020; Yang et al., 2022). Generally, primal-dual methods alternate between the primal update for policy improvement and the dual update for Lagrange multiplier adjustment. While enjoying theoretical guarantee in converging to constraint-satisfying policies (Tessler et al., 2018; Paternain et al., 2019), primal-dual methods often suffer from severe performance instability during the practical training process (Stooke et al., 2020), which can be largely attributed to the high reliance on an accurate estimation of the cumulative cost (Xu et al., 2021; Liu et al., 2022). An inaccurate cost estimation, even with a small error, can give rise to a wrong Lagrange multiplier that hinders the subsequent policy optimization. For example, when the

---

\*Equal contribution. Work done when Bo Tang was at Meituan.
†Corresponding author

true cost value is under the constraint threshold but its estimate is over the threshold (and vice-versa), the Lagrange multiplier will be updated towards the complete opposite of the true direction and in turn becomes a misleading penalty weight in the primal policy objective. Therefore, compared to the reward estimation where an inaccurate value estimate may still be useful for policy improvement as long as the relative reward priority among actions is preserved, cost estimation is more subtle since a minor error can significantly affect the overall primal-dual iteration (Lee et al., 2022b; Liu et al., 2023). This issue is further aggravated when using off-policy methods due to the inherent estimation error caused by the distribution shift (Levine et al., 2020). As a result, the majority of existing primal-dual methods are on-policy by design and thus require a large amount of samples before convergence, which is problematic in safety-critical applications where interacting with the environment could be potentially risky.

In this paper, we propose an off-policy primal-dual-based safe RL method which consists of two main ingredients. The first ingredient is *conservative policy optimization*, which utilizes an upper confidence bound of the true cost value in the Lagrangian, in order to address the cost underestimation issue existing in naïve off-policy primal-dual methods. The underestimation of the cost value causes the policy to violate the constraint, leading to an aggressive feasibility boundary ( ) that exceeds the feasible policy space ( ), as illustrated in Figure 1(a). In contrast, by considering estimation uncertainty, the conservative policy optimization encourages cost overestimation, which ultimately results in a conservative boundary ( ) that lies inside the constraint-satisfying policy space. While such conservatism improves constraint satisfaction, it may also hinder reward maximization since it shrinks the policy search space.

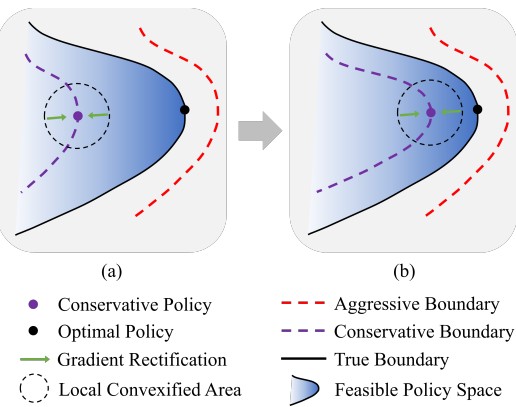

(a)                (b)

- ● Conservative Policy  - - - Aggressive Boundary
- ● Optimal Policy  - - - Conservative Boundary
- → Gradient Rectification  —— True Boundary
- ⬭ Local Convexified Area  ◗ Feasible Policy Space

Figure 1: Visualization of a part of the policy space, where deeper color indicates higher reward in the feasible (constraint-satisfying) area.

To tackle this problem, we introduce our second ingredient, namely *local policy convexification*, which modifies the original objective using the augmented Lagrangian method (Luenberger et al., 1984), in order to convexify the neighborhood area ( ) of a locally optimal policy (●). Through rectifying the policy gradient (→), the local policy convexification can stabilize policy learning in the local convexified area and thus gradually reduce the cost estimation uncertainty in this area. Consequently, the conservative search space will expand gradually towards the optimal policy (●), as depicted by the process from Figure 1(a) to Figure 1(b). The contributions of this work are as follows:

- We analyze the problem existing in naïve off-policy primal-dual methods and address it by introducing two critical algorithmic ingredients. Detailed theoretical interpretations and empirical verifications are provided to demonstrate the respective effect of each ingredient and more importantly, their mutual impact on the overall learning process.

- We specify a practical algorithm by incorporating the two ingredients into an off-policy backbone. Empirical results on various continuous control tasks show that our method achieves superior sample efficiency compared to the baselines, i.e., achieving the same reward with fewer samples and constraint violations during training.

- We evaluate our method on a real-world auto-bidding task under the semi-batch training paradigm (Matsushima et al., 2021), where the behavior policy is not allowed to update within each long-term data collecting process. Results verify the effectiveness of our method in such scenarios by conservatively approaching the optimal policy.

## 2 BACKGROUND

This paper studies safe RL problems under the formulation of the Constrained Markov Decision Process (CMDPs) (Altman, 1999), which is defined by a tuple $(\mathcal{S}, \mathcal{A}, P, r, c, \gamma, d)$, where $\mathcal{S}$ is the state space, $\mathcal{A}$ the action space, $P : \mathcal{S} \times \mathcal{A} \times \mathcal{S} \to [0, 1]$ the transition kernel, $r : \mathcal{S} \times \mathcal{A} \to \mathbb{R}$

the reward function, $c : \mathcal{S} \times \mathcal{A} \to \mathbb{R}^+$ the cost function, $\gamma$ the discount factor, and $d$ the constraint threshold. Without loss of generality, this paper considers CMDPs with only a single scalar cost to simplify the notation. At each time step $t$, the agent selects an action $a_t \in \mathcal{A}$ based on the current state $s_t \in \mathcal{S}$ and the policy $\pi(\cdot|s_t)$. The environment then returns a reward $r_t = r(s_t, a_t)$, a cost $c_t = c(s_t, a_t)$, and the next state $s_{t+1} \in \mathcal{S}$ sampled from the transition probability $P(\cdot|s_t, a_t)$. We denote the reward value function and the cost value function respectively as $Q_r^\pi(s, a) = \mathbb{E}_{\tau \sim \pi}[\sum_{t=0}^\infty \gamma^t r_t]$ and $Q_c^\pi(s, a) = \mathbb{E}_{\tau \sim \pi}[\sum_{t=0}^\infty \gamma^t c_t]$, where $\tau = \{s_0 = s, a_0 = a, s_1, \ldots\}$ is a trajectory starting from $s$ and $a$. The goal of safe RL methods is to solve the following constrained optimization problem:

$$\max_\pi \ \mathbb{E}_{s \sim \rho_\pi, a \sim \pi(\cdot|s)}\left[Q_r^\pi(s, a)\right] \quad \text{s.t. } \mathbb{E}_{s \sim \rho_\pi, a \sim \pi(\cdot|s)}\left[Q_c^\pi(s, a)\right] \leq d, \tag{1}$$

where $\rho_\pi$ denotes the stationary state distribution induced by $\pi$. The basic primal-dual approach converts Eq. (1) to its dual form, i.e.,

$$\min_{\lambda \geq 0} \max_\pi \mathbb{E}_{s \sim \rho_\pi, a \sim \pi(\cdot|s)} \left[Q_r^\pi(s, a) - \lambda\left(Q_c^\pi(s, a) - d\right)\right], \tag{2}$$

and solves it by alternating between optimizing the policy and updating the Lagrange multiplier. For a large state space, the value functions can be estimated with function approximators, and we denote $\widehat{Q}_r^\pi$ and $\widehat{Q}_c^\pi$ as the estimation of the reward and the cost value, respectively.

## 3 METHODOLOGY

This section introduces two key algorithmic ingredients for improving off-policy primal-dual safe RL methods. In Section 3.1, we reveal the cost underestimation issue existing in naïve off-policy primal-dual methods and address it by applying the first ingredient, i.e., conservative policy optimization. To mitigate the possible reward suboptimality induced by such conservatism, we then introduce the second ingredient, i.e., local policy convexification in Section 3.2. The interpretation and empirical verification for the joint effect of these two ingredients are provided in Section 3.3.

### 3.1 CONSERVATIVE POLICY OPTIMIZATION

In the primal-dual objective (2), when the Lagrange multiplier is greater than 0, the policy will be updated to minimize the estimation of the cost value. Assuming that the approximation $\widehat{Q}_c(s, a) = Q_c(s, a) + \epsilon$ has a zero-mean noise $\epsilon$, then the zero-mean property cannot be preserved in general under the minimization operation, i.e., $\mathbb{E}_\epsilon[\min_\pi \widehat{Q}_c(s, \pi(s))] \leq \min_\pi Q_c(s, \pi(s)), \forall s$ (Thrun & Schwartz, 1993), leading to the underestimation of the cost value. Such underestimation bias may be further accumulated to a large bias in temporal difference learning where the underestimated cost value becomes a learning target for estimates of other state-action pairs. This issue becomes even more severe when using off-policy methods that inherently suffer from value approximation error (i.e., larger $\epsilon$) induced by the distribution shift issue (Levine et al., 2020).

We empirically validate the above analysis by showing the results of a naïve off-policy baseline: SAC-Lag, which is the Lagrangian version of SAC (Haarnoja et al., 2018). Figure 2 provides the reward (left column) and the cost evaluation (middle column), as well as the difference between the estimations of cost values and their oracle values during training (right column). The oracle value is approximated by testing the current policy in the environment for 100 episodes and averaging the induced discounted cost returns of these episodes. Clearly, we can see that SAC-Lag suffers from severe underestimation of the cost value and also poor constraint satisfaction.

To deal with such cost underestimation, we propose to replace the value estimate $\widehat{Q}_c^\pi$ in the primal-dual objective by an upper confidence bound (UCB) of the true cost value. Specifically, we maintain $E$ bootstrapped $Q_c$ networks, i.e., $\{\widehat{Q}_c^i\}_{i=1}^E$, which share the same architecture and only differ by the initial weights. All the networks are trained on the same dataset. During cost estimation, each $\widehat{Q}_c^i$ estimates the cumulative cost in parallel and independently. The UCB value is obtained by computing "mean plus weighted standard deviation" of this bootstrapped value ensemble:

$$\widehat{Q}_c^{\text{UCB}}(s, a) = \frac{1}{E}\sum_{i=1}^E \widehat{Q}_c^i(s, a) + k \cdot \sqrt{\frac{1}{E}\sum_{i=1}^E \left(\widehat{Q}_c^i(s, a) - \frac{1}{E}\sum_{i=1}^E \widehat{Q}_c^i(s, a)\right)^2}, \tag{3}$$

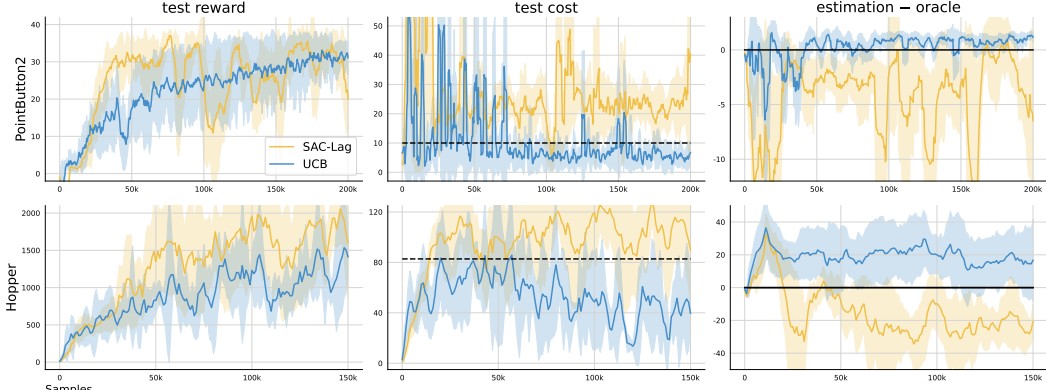

Figure 2: Results of SAC-Lag and its conservative version (denoted as "UCB" in the legend) on two example tasks. The horizontal dashed line represents the constraint threshold in the cost plot.

where we omit the superscript $\pi$ in $\widehat{Q}_c^{\text{UCB}}$ and $\widehat{Q}_c^i$ for notational brevity. From the Bayesian perspective, the ensemble forms an estimation of the posterior distribution of $Q_c$ (Osband et al., 2016), with the standard deviation quantifying the epistemic uncertainty induced by a lack of sufficient data (Chua et al., 2018). Despite its simplicity, this technique is widely adopted and is proved to be extremely effective (Ghasemipour et al., 2022; Bai et al., 2022). By taking into account this uncertainty in $Q_c$ estimation, $\widehat{Q}_c^{\text{UCB}}$ serves as an approximate upper bound of the true value with high confidence, as empirically verified by Figure 2 [1]. Intuitively, the use of UCB induces safety conservativeness in the policy by encouraging actions that not only result in high reward but also yield low uncertainty in cost estimation.

## 3.2 LOCAL POLICY CONVEXIFICATION

While conservative policy optimization can improve constraint satisfaction by encouraging cost overestimation, it could also impede reward maximization due to the shrink of policy search space (i.e., $\mathbb{E}[Q_c^\pi] \leq \mathbb{E}[\widehat{Q}_c^{\text{UCB}}]$ implies $\{\pi | \mathbb{E}[\widehat{Q}_c^{\text{UCB}}] \leq d\} \subseteq \{\pi | \mathbb{E}[Q_c^\pi] \leq d\}$). As illustrated in Figure 1, by narrowing the gap between the conservative boundary ($\odot$) and the true boundary ($\triangleright$), the algorithm can expand the search space and thus enhance the probability to obtain a high-reward policy. However, this conservatism gap is co-controlled by the conservatism parameter $k$ and the bootstrap disagreement. Since the bootstrap disagreement quantifies the uncertainty and is susceptible to instability, it is impractical to precisely control the gap only through tuning the conservatism parameter. In this section, we introduce local policy convexification to stabilize the learning of the policy and the Lagrange multiplier. When working in conjunction with conservative policy optimization, such stabilization helps reduce the bootstrap disagreement and thus gradually narrows down the conservatism gap, as interpreted in Section 3.3.

In the estimation of the reward value, even if the estimated values are far from their true values, they can still be helpful for policy improvement as long as their relative order is preserved. Nevertheless, the cost value estimation is less tolerant to estimation error, because the dual update process needs to compare its exact value against an absolute threshold. Thus, even a slight estimation error of the cost value can result in a completely wrong $\lambda$ that misleads the subsequent primal optimization. For instance, in objective (2) the optimal value for $\lambda$ is $\infty$ if the estimated cost value exceeds the threshold, and is $0$ otherwise. Therefore, compared to on-policy methods, off-policy methods are less suitable in primal-dual safe RL due to the inherent value estimation error (Levine et al., 2020), which can lead to instability in policy learning and oscillations in the overall performance. To stabilize the training of off-policy methods, we modify the original objective (2) by applying the augmented Lagrangian method (ALM) (see Pages 445-453 of (Luenberger et al., 1984)) as follows [2]:

---

[1] For presentational convenience, we only show the performance of $k = 2$ in Figure 2, and the ablation study of this hyperparameter is provided in Section 3.3.

[2] For the convenience of the following interpretations, here we use the UCB modified objective by replacing $\widehat{Q}_c^\pi$ in the original objective (2) with $\widehat{Q}_c^{\text{UCB}}$.

$$\begin{cases} \min_{\lambda \geq 0} \max_{\pi} \mathbb{E}\left[\widehat{Q}_r^{\pi}\right] - \lambda \left(\mathbb{E}\left[\widehat{Q}_c^{\text{UCB}}\right] - d\right) - \frac{c}{2}\left(\mathbb{E}\left[\widehat{Q}_c^{\text{UCB}}\right] - d\right)^2, & \text{if } \frac{\lambda}{c} > d - \mathbb{E}[\widehat{Q}_c^{\text{UCB}}]; \quad (4a) \\ \max_{\pi} \mathbb{E}\left[\widehat{Q}_r^{\pi}\right], & \text{otherwise,} \quad\quad\quad (4b) \end{cases}$$

where $c$ is a positive hyperparameter and $s \sim \rho_{\pi}, a \sim \pi(\cdot|s)$. Such a modification does not alter the positions of local optima under a common assumption for inequality constraints (see Appendix A). Note that (4b) does not involve the cost value, which is reasonable since $\frac{\lambda}{c} \leq d - \mathbb{E}[\widehat{Q}_c^{\text{UCB}}]$ implies that the UCB value is below the threshold. As for objective (4a), intuitively, with a sufficiently large $c$, it will be dominated by the quadratic penalty term and thus will be convex in the neighborhood of a boundary solution (i.e., a policy $\pi$ satisfying $\mathbb{E}[\widehat{Q}_c^{\text{UCB}}] = d$). Theoretically, assuming a continuous-time setting, we can define the "energy" of our learning system as $\mathbf{E} = \|\dot{\pi}\|^2 + \frac{1}{2}\dot{\lambda}^2$, where the dot denotes the gradient w.r.t. time. Adapting the proof in (Platt & Barr, 1987) to the inequality constraint case leads to the following proposition (see Appendix B for the complete proof):

**Proposition 1.** *If $c$ is sufficiently large, $\mathbf{E}$ will decrease monotonically during training in the neighborhood of a local optimal policy.*

*Proof (Sketch).* Let $f(\pi) = \mathbb{E}_{s,a}[\widehat{Q}_r^{\pi}(s,a)]$ and $g(\pi) = \mathbb{E}_{s,a}[\widehat{Q}_c^{\text{UCB}}(s,a)] - d$. By transforming the objective into an equivalent form and using the chain rule, the dynamics of the energy $\mathbf{E}$ can then be derived as: $\dot{\mathbf{E}} = -\dot{\pi}^T A \dot{\pi}$, where $u \geq 0$ and $A = -\nabla^2 f + (\lambda + c(g(\pi) + u))\nabla^2 g + c(\nabla g)^2$. According to Lemma 1, there exists a $c^* > 0$ such that if $c > c^*$, $A$ is positive definite at the constrained minima. Using continuity, $A$ is positive definite in some neighborhoods of local optima, hence, the energy function $E$ monotonically decreases in such neighborhood areas. $\square$

By definition, a relatively smaller $\mathbf{E}$ implies smaller update steps of both $\pi$ and $\lambda$, meaning that the policy will oscillate less when it is close to a local minimum. In other words, there will be an "absorbing area" surrounding each suboptimal policy, in which the updating policy can be stabilized and thus be attracted by this local optimum.

### 3.3 JOINT EFFECT OF CONSERVATIVE POLICY OPTIMIZATION AND LOCAL POLICY CONVEXIFICATION

Differentiating the ALM modified objective (4a) w.r.t. the estimated cost value yields the gradient $\mathbb{E}[\tilde{\lambda}\nabla_a \widehat{Q}_c^{\text{UCB}}(s,a)|_{a=\pi_{\phi}(s)}\nabla_{\phi}\pi_{\phi}(s)]$, where $\tilde{\lambda} = \max\{0, \lambda - c(d - \mathbb{E}[\widehat{Q}_c^{\text{UCB}}(s,a)])\}$. Therefore, the ALM modification can be seen as applying a **gradient rectification** by replacing $\lambda$ in the original gradient with $\tilde{\lambda}$. As illustrated in Figure 1, the effect of such gradient rectification ($\rightarrow$) is manifested in two cases:

1) when the policy crosses the conservative boundary (i.e., $\mathbb{E}[\widehat{Q}_c^{\text{UCB}}(s,a)] > d$), the ALM modification rectifies the policy gradient by assigning a larger penalty weight (i.e., $\lambda - c(d - \mathbb{E}[\widehat{Q}_c^{\text{UCB}}(s,a)]) > \lambda$), which is an acceleration for dragging the policy back to the conservative area; 2) in contrast, when the policy satisfies the conservative constraint (i.e., $\mathbb{E}[\widehat{Q}_c^{\text{UCB}}(s,a)] \leq d$) while $\lambda$ is still large and not able to be immediately reduced to 0, we have $\lambda - c(d - \mathbb{E}[\widehat{Q}_c^{\text{UCB}}(s,a)]) \leq \lambda$, which to some extent prevents unnecessary over-conservative policy updates from reducing the cost value.

Therefore, applying local policy convexification can force the policy to stay close to the local optimum, which is consistent with the theoretical analysis from the energy perspective. With such a stabilization effect, the collected samples can be focused in line with the distribution of the local optimal policy, which in turn dissipates the epistemic uncertainty in the local convexified area ($\bigcirc$ in Figure 1). As the uncertainty decreases, the value estimations in the ensemble ($\{\widehat{Q}_c^i\}_{i=1}^E$ in Eq. (3) will become more accurate, i.e.,

$$\frac{1}{E}\sum_{i=1}^{E}\widehat{Q}_c^i(s,a) \rightarrow Q_c^{\pi}(s,a), Std_{i=1,\dots,E}\left(\widehat{Q}_c^i(s,a)\right) \rightarrow 0 \quad (5)$$

and the bootstrap disagreement, which quantifies the epistemic uncertainty, will also decrease, i.e.,

$$\sqrt{\frac{1}{E}\sum_{i=1}^{E}\left(\widehat{Q}_c^i(s,a) - \frac{1}{E}\sum_{i=1}^{E}\widehat{Q}_c^i(s,a)\right)^2} \rightarrow 0. \quad (6)$$

In this way, the conservativeness induced by the bootstrap disagreement is reduced, and thus the conservative boundary can be pushed gradually towards the true boundary.

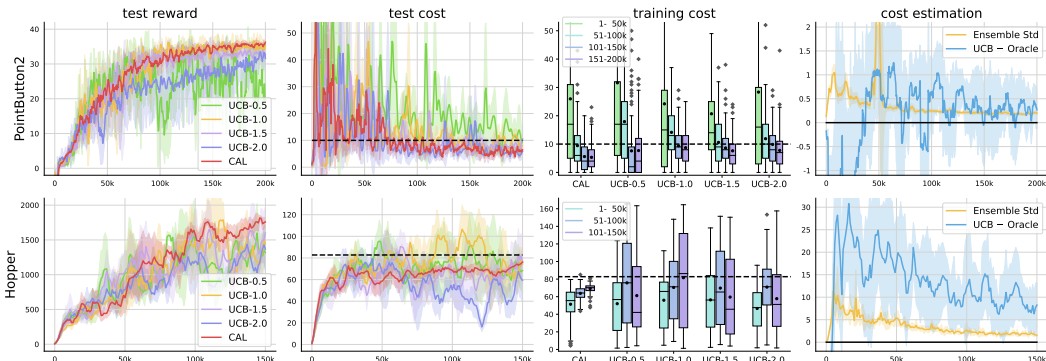

Figure 3: Results of applying local policy convexification on two example tasks, averaged over 6 random seeds. **The first two columns** show the reward/cost learning curves, where the solid lines are the mean and the shadowed regions are the standard deviation. **The third column** shows the training cost of different timestep intervals by using box-and-whisker plots where boxes showing median, $75\%$ ($q_3$) and $25\%$ ($q_1$) quarterlies of the distributions respectively, whiskers depicting the error bounds computed as $1.5(q_3 - q_1)$, as well as outliers lying outside the whisker intervals. **The last column** shows the trend of the standard deviation of the cost value ensemble during training, as well as the difference between the UCB cost value and its oracle value during training (where greater than 0 implies overestimation, otherwise implies underestimation).

To validate the joint effect of the two introduced ingredients, we incorporate them into the off-policy RL method of SAC and denote the final algorithm as Conservative Augmented Lagrangian (CAL) [3]. The results are given in Figure 3, where "UCB-$k$" denotes the method only using UCB with $k$ being the value of the conservatism parameter, and $k$ in CAL is set to 0.5 in the two tested tasks. From **the first two columns**, we can see that compared to the results of merely depending on the conservative policy optimization, applying local policy convexification yields significantly smaller oscillations, indicating that this algorithmic ingredient can stabilize the overall performance both within a single run (the solid line) and across different runs (the shadowed region). Moreover, **the third column** [4] shows that, 1) as the timestep grows, the episodic training cost becomes more and more stable (embodied in the decreasing box length) and closer to the constraint thresholds; and 2) even compared to sole UCB with a much larger $k$, using local policy convexification makes the overall constraint violations and cost oscillations much fewer during training. Additionally, we can see in **the last column** that the overestimated UCB value gradually approaches its oracle value while the standard deviation of the bootstrapped ensemble decays toward 0, which is consistent with our interpretation about the coupling effect of the two ingredients [5].

## 4  EXPERIMENTS

In this section, we first compare CAL to several strong baselines on benchmarks in Section 4.1, and then study two hyperparameters that respectively control the effect of the two proposed ingredients in Section 4.2. The experiment on a real-world advertising bidding scenario is presented in Section 4.3.

---

[3]The pseudo code of CAL can be found in Appendix C.

[4]Note that the learning curves are the testing results using extra evaluation episodes after each training iteration, while the box plots presents the training cost.

[5]Note that different from the results on Hopper, the UCB value in the PointButton2 task does not exhibit overestimation at the early stage of training. This is because the sparsity of the cost in the PointButton2 task increases the difficulty in learning the value estimates.

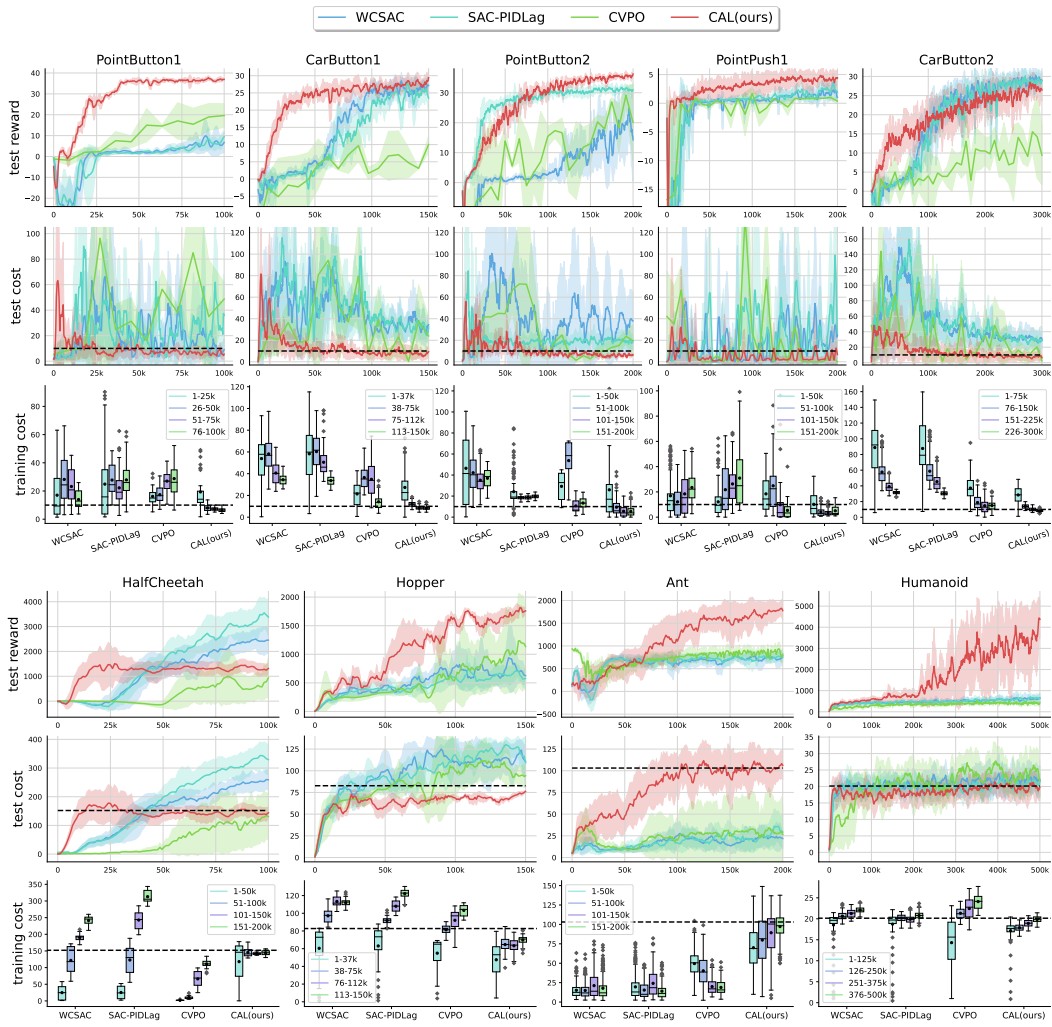

Figure 4: Comparisons with off-policy baselines on Safety-Gym (top half) and velocity-constrained MuJoCo (bottom half).

## 4.1 COMPARATIVE EVALUATION

We conduct our comparative evaluation on the Safety-Gym benchmark (Ray et al., 2019) and the velocity-constrained MuJoCo benchmark (Zhang et al., 2020). A detailed description of the tested tasks can be found in Appendix D.1. Baselines are categorized into off- and on-policy methods. The off-policy baselines include 1) SAC-PIDLag (Stooke et al., 2020), which builds a stronger version of SAC-Lag by using PID-control to update the Lagrange multiplier, 2) WCSAC (Yang et al., 2021), which replaces the expected cost value in SAC-Lag with the Conditional Value-at-Risk (Rockafellar et al., 2000) to achieve controllable conservatism, and 3) CVPO (Liu et al., 2022), which performs constrained policy learning in an Expectation Maximization fashion. Due to the space limit, we show the comparisons with on-policy baselines in Appendix D.2. Implementation details of CAL and the baselines are given in Appendix D.3. Our code can be found in our supplementary material.

Figures 4 shows the results on the benchmarks averaged over 6 random seeds. The top/middle rows present the tested reward/cost using extra episodes after each training iteration [6]. The bottom row presents the cost induced during training, where the training episodes are quartered in time order and respectively represented by the four boxes for each algorithm. The horizontal dashed lines in the

---

[6]Note that the smoothness of the learning curves among the baselines varies due to the different testing intervals, which result from the difference in the training paradigm.

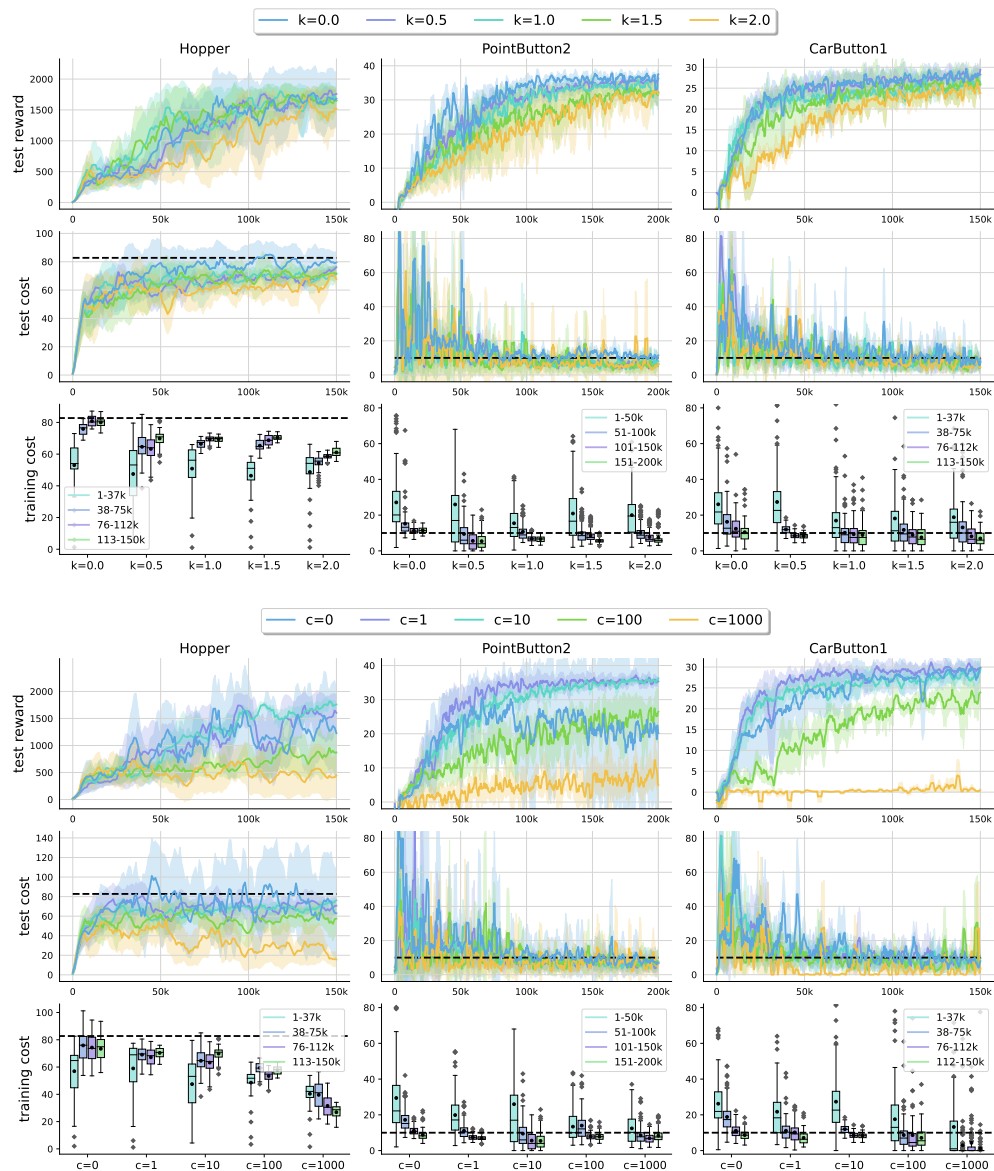

Figure 5: Ablation Studies of $k$ (top half) and $c$ (bottom half).

cost plots represent the constraint thresholds. On all tested tasks, CAL significantly outperforms all baselines in low data regime, and also achieves the same level of reward and constraint satisfaction as that of state-of-the-art on-policy methods (see Appendix D.2), with much fewer samples and constraint violations during training.

## 4.2 ABLATION STUDIES

An appealing advantage of CAL is that there are only a small number of hyperparameters that need to be tuned. The ablation studies for two key hyperparameters are presented in Figure 5, i.e., $k$ in Eq. (3) and $c$ in Eq. (4a), which respectively control the conservativeness in conservative policy optimization and the degree of convexity in local policy convexification. As shown in the top half of Figure 5, a larger $k$ generally leads to a lower test/training cost, but an overly conservative $k$ can also hinder reward maximization. As a special case, setting $k = 0$ leads to significantly more constraint violations, and it even fails to obtain an asymptotic constraint-satisfying policy on PointButton2 and CarButton1, which demonstrates the importance of incorporating conservatism into the algorithm. As for the ablation results of $c$, increasing $c$ results in fewer performance oscillations but slower

reward improvement. Specially, setting $c = 0$ (i.e., without ALM) results in severe performance oscillations which hinder both the reward maximization and constraint satisfaction, and an overly large $c$ imposes a large penalty on policies exceeding the conservative boundary (where $\mathbb{E}[\widehat{Q}_c^{\text{UCB}}] > d$), which significantly slows down the process of expanding the conservative boundary (as illustrated in Figure 1) and thus impedes reward improvement.

In addition, an ablation study for the ensemble size $E$ is shown in Figure 8 in Appendix D.4, where we can see that a larger $E$ generally leads to a lower cost, but the overall performance is not very sensitive to this hyperparameter. In theory, increasing the number of $Q_c$ networks will improve the reliability of uncertainty quantification. However, Figure 8 shows that in simpler tasks (i.e., Hopper and PointButton2) the benefit of doing this is not as significant as in the harder CarButton2. Since training more networks yields more computational burden, users of the CAL algorithm will need to make some trade-offs according to the difficulty of the target task and the computational expenses.

## 4.3 SEMI-BATCH SETTING

Semi-batch setting refers to the training paradigm where the agent alternates between 1) collecting a large batch of data with a fixed behavior policy, and 2) updating the target policy on the collected data. Since the policy is not allowed to be adjusted during long-term environment interaction, the online deployment of a policy must ensure a certain level of safety. As a compromise between the online and the offline training (Levine et al., 2020), such a training paradigm is crucial for real-world applications of RL (Lange et al., 2012; Matsushima et al., 2021). In this regard, CAL may be particularly useful since by conservatively approaching the optimal policy from the constraint-satisfying area it can ensure both safe online deployment and high offline sample efficiency.

To verify CAL's capability in the semi-batch setting, we conduct experiments in a large-scale advertising bidding scenario w.r.t. oCPM (the optimized cost-per-mille), which is a typical semi-batch application in real world. In the advertising bidding scenario, advertisers bid for advertisement slots that would be exhibited to users browsing the site. After being assigned the target ROI (return on investment) of each advertiser, the auto-bidding policy provided by the platform helps the advertisers bid for the slots, in order to maximize the total payments of all advertisers while satisfying each ROI constraint. A more detailed description of this scenario is provided in Appendix D.1.

Our experiment is carried out on a real-world advertising platform with a total amount of 204.8 million bidding records collected from 47244 advertisers within 4 weeks. During the online deployment of each week, the platform generates 51.2 million records with a fixed bidding policy. The policy is only allowed to update once a week on data sampled from the historical dataset. We evaluate the algorithms by summarizing the total advertiser payments and the ROI violation rate within each deployment. The experiment results shown in Table 1 demonstrate that CAL significantly outperforms the baselines in terms of payment maximization while maintaining relatively low ROI violation rate, verifying CAL's potential to the large-scale semi-batch setting on real-world applications.

|  | WCSAC | | CVPO | | CUP | | SAC-PIDLag | | **CAL(ours)** | |
|---|---|---|---|---|---|---|---|---|---|---|
|  | Pay$/k$ | Vio/% | Pay$/k$ | Vio/% | Pay$/k$ | Vio/% | Pay$/k$ | Vio/% | Pay$/k$ | Vio/% |
| Week 1 | 270.23 | **30.43** | 374.38 | 31.66 | 433.33 | 31.86 | 367.82 | 32.30 | **656.31** | 30.55 |
| Week 2 | 96.38 | 25.51 | 92.23 | 23.44 | 116.89 | **22.63** | 105.12 | 25.77 | **372.51** | 23.04 |
| Week 3 | 0 | 22.57 | 25.20 | 21.38 | 59.47 | 20.88 | 29.71 | 23.12 | **208.65** | **18.15** |
| Week 4 | 25.61 | 22.70 | 51.06 | 20.52 | 95.80 | 19.50 | 58.22 | 23.07 | **226.37** | **17.26** |

Table 1: Comparison on the oCPM task, where "Pay" refers to the total advertiser payments (value normalized for data security) and "Vio" refers to the ROI violation rate.

## 5 CONCLUSION

This paper proposes an off-policy primal-dual method containing two main ingredients, i.e., conservative policy optimization and local policy convexification. We provide both theoretical interpretation and empirical verification of how these two ingredients work in conjunction with each other. In addition to the benchmark results which demonstrate a significantly higher sample efficiency compared to the baselines, the proposed method also exhibits promising performance on a large-scale real-world task using a challenging semi-batch training paradigm. A possible direction for future work is to extend our approach to fully offline settings.

ACKNOWLEDGMENTS

We gratefully acknowledge support from the National Natural Science Foundation of China (No. 62076259), the Fundamental and Applicational Research Funds of Guangdong province (No. 2023A1515012946), and the Fundamental Research Funds for the Central Universities-Sun Yat-sen University. This research is also supported by Meituan.

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

# A  ASSUMPTIONS AND PROOF FOR "THE ALM MODIFICATION FOR INEQUALITY-CONSTRAINT PROBLEMS DOES NOT ALTER THE POSITIONS OF LOCAL OPTIMA"

The ALM modification in Section 3.2 does not alter the positions of local optima under the following strict complementarity assumption (Luenberger et al., 1984):

**Assumption 1** (Strict Complementarity). *$\mathbb{E}_{\pi^*}[Q_c^{\pi^*}(s,a)] = d$ implies that $\lambda^* > 0$ as well as the **second order sufficiency conditions** hold for the original constrained optimization problem, where $\pi^*$ denotes the optimal solution to the original constrained problem, and $\lambda^*$ denotes the optimal solution to the dual problem.*

**Assumption 2** (Second Order Sufficiency). *The second order sufficiency conditions for the original constrained optimization problem include:*

- $\mathbb{E}_{\pi^*}[Q_c^{\pi^*}(s,a)] = d$

- *there exists $\lambda \geq 0$, s.t.*

$$\nabla\mathbb{E}_{\pi^*}[Q_r^{\pi^*}(s,a)] + \lambda\nabla\mathbb{E}_{\pi^*}[Q_c^{\pi^*}(s,a)] = 0 \tag{7}$$

- *the Hessian matrix*

$$\nabla^2\mathbb{E}_{\pi^*}[Q_r^{\pi^*}(s,a)] + \lambda\nabla^2\mathbb{E}_{\pi^*}[Q_c^{\pi^*}(s,a)] \tag{8}$$

  *is positive definite on the subspace $\{\pi|\nabla\pi^*\mathbb{E}_{\pi^*}[Q_c^{\pi^*}(s,a)] = 0\}$.*

Here we prove that *under the strict complementarity assumption, the ALM modification does not alter the position of local optima.*

*Proof.* Let $f(\pi) = \mathbb{E}_{s\sim\rho_\pi, a\sim\pi(\cdot|s)}\left[\widehat{Q}_r^\pi(s,a)\right]$ and $g(\pi) = \mathbb{E}_{s\sim\rho_\pi, a\sim\pi(\cdot|s)}\left[\widehat{Q}_c^{\text{UCB}}(s,a)\right] - d$, the original problem can be written as

$$\max_\pi \ f(\pi) \quad \text{s.t. } g(\pi) \leq 0. \tag{9}$$

If the strict complementarity assumption is satisfied, then the original problem can be written as an equivalent problem with equality constraints (Luenberger et al., 1984):

$$\begin{aligned} \max_\pi \ & f(\pi) \\ \text{s.t. } & g(\pi) + u = 0, \quad u \geq 0. \end{aligned} \tag{10}$$

Since $c > 0$, the optimization problem (10) is equivalent to:

$$\begin{aligned} \max_\pi \ & f(\pi) - \frac{1}{2}c|g(\pi) + u|^2 \\ \text{s.t. } & g(\pi) + u = 0, \quad u \geq 0, \end{aligned} \tag{11}$$

which does not alter the position of local optima since the conditions of stationary points of the Lagrangian of Eq. (11) are not changed compared to the original Lagrangian. Then, we formulate the dual function of problem (11) as:

$$\phi(\lambda) = \max_{u\geq 0, \pi} f(\pi) - \lambda^T[g(\pi) + u] - \frac{1}{2}c|g(\pi) + u|^2, \tag{12}$$

where $\lambda$ is the Lagrangian multiplier. Now we show that the optimization of the dual problem, i.e., $\min_{\lambda\geq 0} \phi(\lambda)$, is equivalent to:

$$\min_{\lambda\geq 0} \max_\pi \mathbb{E}\left[\widehat{Q}_r^\pi\right] - \frac{1}{2c}\left[\left(\max\{0, \lambda - c(d - \mathbb{E}\left[\widehat{Q}_c^{\text{UCB}}\right]\}\right)^2 - \lambda^2\right]. \tag{13}$$

Given $\lambda$ and $\pi$, the analytical optimal solution $u_j^*$ for problem (12) can be obtain by solving the quadratic problem:

$$P_j = \max_{u_j} -\frac{1}{2}c[g_j(\pi) + u_j]^2 - \lambda_j[g_j(\pi) + u_j] \tag{14}$$

It can be easily computed that the derivative of Eq. (14) is 0 at $-g_j(\pi) - \lambda_j/c$. Thus, the optimal solution to problem (12) is $u_j^* = 0$ if $-g_j(\pi) - \lambda_j/c < 0$, otherwise $u_j^* = -g_j(\pi) - \lambda_j/c$. In other words, the solution can be written as:

$$u_j^* = \max\{0, -g_j(\pi) - \lambda_j/c\}. \tag{15}$$

Substituting Eq. (15) into Eq. (14), we obtain an explicit expression for $P_j$: 1) for $u_j^* = 0$, we have:

$$P_j = -\frac{1}{2}c \cdot g_j(\pi)^2 - \lambda_j \cdot g_j(\pi) = -\frac{1}{2c}\left([\lambda_j + c \cdot g_j(\pi)]^2 - \lambda_j^2\right); \tag{16}$$

and 2) for $u_j^* = -g_j(\pi) - \lambda_j/c$, we have:

$$P_j = \lambda_j^2/2c. \tag{17}$$

Combining the two cases together we get:

$$P_j = -\frac{1}{2c}\left([\max\{0, \lambda_j + c \cdot g_j(\pi)\}]^2 - \lambda_j^2\right). \tag{18}$$

Therefore, the dual function can be written as the following form that only depends on $\lambda$ and $\pi$:

$$
\begin{aligned}
\phi(\lambda) &= \max_{u \geq 0, \pi} f(\pi) - \lambda^T[g(\pi) + u] - \frac{1}{2}c|g(\pi) + u|^2 \\
&= \max_{\pi} f(\pi) + \sum_j P_j \\
&= \max_{\pi} f(\pi) - \frac{1}{2c}\left(|\max\{0, \lambda + c \cdot g(\pi)\}|^2 - |\lambda|^2\right) \\
&= \max_{\pi} \mathbb{E}\left[\widehat{Q}_r^\pi\right] - \frac{1}{2c}\left[\left(\max\{0, \lambda - c(d - \mathbb{E}\left[\widehat{Q}_c^{\mathrm{UCB}}\right]\}\right)^2 - \lambda^2\right].
\end{aligned} \tag{19}
$$

According the equality in Eq. (19), the optimization of the dual problem (12) $\min_{\lambda \geq 0} \phi(\lambda)$ is equivalent to Eq. (13). Based on this result, the objective (4a)(4b) in the main paper can be derived, i.e.,

$$
\min_{\lambda \geq 0} \phi(\lambda) = 
\begin{cases}
\min\limits_{\lambda \geq 0} \max\limits_{\pi} \mathbb{E}\left[\widehat{Q}_r^\pi\right] - \lambda\left(\mathbb{E}\left[\widehat{Q}_c^{\mathrm{UCB}}\right] - d\right) - \frac{c}{2}\left(\mathbb{E}\left[\widehat{Q}_c^{\mathrm{UCB}}\right] - d\right)^2, & \text{if } \frac{\lambda}{c} > d - \mathbb{E}[\widehat{Q}_c^{\mathrm{UCB}}] \\[2mm]
\min\limits_{\lambda \geq 0} \max\limits_{\pi} \mathbb{E}\left[\widehat{Q}_r^\pi\right] + \frac{\lambda^2}{2c}, & \text{otherwise.}
\end{cases}
$$

Since $\min\limits_{\lambda \geq 0} \max\limits_{\pi} \mathbb{E}\left[\widehat{Q}_r^\pi\right] + \frac{\lambda^2}{2c}$ is equivalent to $\max\limits_{\pi} \mathbb{E}\left[\widehat{Q}_r^\pi\right]$, we have reached our conclusion that the ALM modification does not alter the positions of local optima. $\qquad \square$

## B  PROOF FOR THE MONOTONIC ENERGY DECREASE AROUND A LOCAL OPTIMUM

In Section 3.2, we define the "energy" of the learning system of Eq. (13) as: $\mathbf{E} = \|\dot\pi\|^2 + \frac{1}{2}\dot\lambda^2$, where the dot denotes the gradient w.r.t. time (assuming a continuous-time setting). Now we extend the proof in (Platt & Barr, 1987) to show that:

*If $c$ is sufficiently large, $\mathbf{E}$ will decrease monotonically during training in the neighborhood of a local optimal policy.*

*Proof.* Recall that in the proof in Appendix A, the equivalent problem for the equality constrained optimization (10) can be written as:

$$
\begin{aligned}
\min_{\pi} \quad & -f(\pi) + \frac{1}{2}c|g(\pi) + u|^2 \\
\text{s.t.} \quad & g(\pi) + u = 0, \quad u \geq 0.
\end{aligned} \tag{21}
$$

The Lagrangian of this constrained objective is:

$$L(\pi, \lambda) = -f(\pi) + \lambda(g(\pi) + u) + \frac{1}{2}c|g(\pi) + u|^2. \tag{22}$$

The continuous learning dynamics of Eq. (22) can be written as:

$$\dot{\pi} = -\frac{\partial L}{\partial \pi} = \frac{\partial f}{\partial \pi} - \lambda \frac{\partial g}{\partial \pi} - c(g(\pi) + u)\frac{\partial g}{\partial \pi}$$
$$\dot{\lambda} = -(g(\pi) + u). \tag{23}$$

During each time interval $\Delta t$, the policy $\pi$ and the Lagrangian multiplier $\lambda$ are updated to $\pi + \dot{\pi}\Delta t$ and $\lambda + \dot{\lambda}\Delta t$, respectively. The dynamics of the energy $\mathbf{E} = \|\dot{\pi}\|^2 + \frac{1}{2}\dot{\lambda}^2$ can be derived as:

$$\dot{\mathbf{E}} = \frac{d}{dt}\left[\frac{1}{2}\sum_i (\dot{\pi}_i)^2 + \frac{1}{2}(g(\pi) + u)^2\right]$$
$$= \sum_i \frac{d}{dt}\left[\frac{1}{2}(\dot{\pi}_i)^2\right] + (g(\pi) + u)(\nabla g)^T \dot{x}. \tag{24}$$

Furthermore, since

$$\frac{d}{dt}\left[\frac{1}{2}(\dot{\pi}_i)^2\right] = \dot{\pi}_i \frac{d}{dt}\left[\frac{\partial f}{\partial \pi_i} - \lambda \frac{\partial g}{\partial \pi_i} - c(g(\pi) + u)\frac{\partial g}{\partial \pi_i}\right]$$
$$= \dot{\pi}_i \left[\sum_j \left[\frac{\partial^2 f}{\partial \pi_i \partial \pi_j} - \lambda \frac{\partial^2 g}{\partial \pi_i \partial \pi_j} - c\frac{\partial g}{\partial \pi_i}\frac{\partial g}{\partial \pi_j} - c(g(\pi) + u)\frac{\partial^2 g}{\partial \pi_i \partial \pi_j}\right]\dot{\pi}_j - \dot{\lambda}\frac{\partial g}{\partial \pi_i}\right], \tag{25}$$

we derive the following expression for $\dot{\mathbf{E}}$ by substituting Eq. (25) into Eq. (24):

$$\dot{\mathbf{E}} = \sum_i \sum_j \dot{\pi}_i \left[\frac{\partial^2 f}{\partial \pi_i \partial \pi_j} - \lambda \frac{\partial^2 g}{\partial \pi_i \partial \pi_j} - c\frac{\partial g}{\partial \pi_i}\frac{\partial g}{\partial \pi_j} - c(g(\pi) + u)\frac{\partial^2 g}{\partial \pi_i \partial \pi_j}\right]\dot{\pi}_j$$
$$- \dot{\lambda}\frac{\partial g}{\partial \pi_i} + (g(\pi) + u)(\nabla g)^T \dot{x}$$
$$= -\dot{\pi}^T \left[-\nabla^2 f + (\lambda + c(g(\pi) + u))\nabla^2 g + c\nabla g(\nabla g)^T\right]\dot{\pi} - (g(\pi) + u)(\nabla g)^T \dot{x} + (g(\pi) + u)(\nabla g)^T \dot{x}$$
$$= -\dot{\pi}^T A\dot{\pi}, \tag{26}$$

where $A = -\nabla^2 f + (\lambda + c(g(\pi) + u))\nabla^2 g + c\nabla g(\nabla g)^T$. Before discussing this damping matrix $A$, we first introduce a Lemma mentioned in (Duguid, 1960) [Page 172]:

**Lemma 1.** *If $P$ is a symmetric matrix and if $x'Px < 0$ for all $x \neq 0$ satisfying $Qx = 0$, then for all $c$ sufficiently large, $P - cQ^T Q$ is negative definite.*

Denoting $\varphi(\pi, \mu) = -f(\pi) + \mu(g(\pi) + u)$ as the Lagrangian function of the original constrained problem (10) and $\pi^*$ as an optimal solution for the original constrained problem, we can easily derive that $\pi^*$ satisfies: 1) $g(\pi^*) + u = 0$, and 2) there exists critical point $(\pi^*, \mu^*)$ such that $\nabla^2\varphi(\pi^*, \mu^*) = -\nabla^2 f(\pi^*) + \mu^*\nabla^2 g(\pi^*)$ is positive definite according to the second order sufficiency conditions. At the point $(\pi^*, \mu^*)$, the damping matrix can be represented by:

$$A^* = -\nabla^2 f(\pi^*) + \mu^*\nabla^2 g(\pi^*) + c\nabla g(\pi^*)(\nabla g(\pi^*))^T$$
$$= -(P - cQ^T Q), \tag{27}$$

where $P = -(-\nabla^2 f(\pi^*) + \mu^*\nabla^2 g(\pi^*))$ and $Q = \nabla g(\pi^*)$. Since $P$ is negative definite, we can apply Lemma 1 and obtain that there exists a $c^* > 0$ such that if $c > c^*$, the damping matrix $A^*$ is positive definite at the constrained minima $(\pi^*, \mu^*)$. Using continuity, the damping matrix $A^*$ is positive definite in some neighborhoods of the local optima, hence, the energy function $\mathbf{E}$ monotonically decreases in such neighborhood areas. $\square$

## C  PSEUDO CODE

The pseudo code of CAL is presented in Algorithm 1.

---

**Algorithm 1** CAL

---

1: Set hyperparameters: $k, c$, and learning rates $\alpha_\pi, \alpha_Q, \alpha_\lambda, \tau$.

2: Initialize policy $\pi_\phi$, double reward Q-functions $\widehat{Q}_r^{\theta_1}, \widehat{Q}_r^{\theta_2}$, cost Q-function ensemble $\{\widehat{Q}_c^{\psi_i}\}_{i=1}^E$, target networks $\widehat{Q}_r^{\theta_1^-}, \widehat{Q}_r^{\theta_2^-}, \{\widehat{Q}_c^{\psi_i^-}\}_{i=1}^E$, Lagrange multiplier $\lambda$, and replay buffer $\mathcal{D}$.

3: **for** each iteration **do**

4:     **for** each environment step **do**

5:         $a_t \sim \pi_\phi(\cdot|s_t)$

6:         $s_{t+1} \sim P(\cdot|s_t, a_t)$

7:         $\mathcal{D} \leftarrow \mathcal{D} \cup \{s_t, a_t, r(s_t, a_t), c(s_t, a_t), s_{t+1}\}$

8:     **end for**

9:     **for** each gradient step **do**

10:        sample experience batch $\mathcal{B} = \{s_{(k)}, a_{(k)}, s'_{(k)}\}_{k=1}^B$ from $\mathcal{D}$

11:        $\phi \leftarrow \phi - \alpha_\pi \mathbb{E}_{s,a\sim\mathcal{B}}\Big[\underbrace{\nabla_\phi \log \pi_\phi(a|s) + \Big(\nabla_a \log \pi_\phi(a|s) - \nabla_a \min_{i\in\{1,2\}} \widehat{Q}_r^{\theta_i}(s,a)\Big)\nabla_\phi \pi_\phi(a|s)}_{\text{SAC objective}}$

12:        $+ \tilde{\lambda}\nabla_a \widehat{Q}_c^{\text{UCB}}(s,a)\nabla_\phi \pi_\phi(a|s)\Big]$, where $\tilde{\lambda} = \max\Big\{0, \lambda - c\Big(d - \mathbb{E}_{s,a\sim\mathcal{B}}\Big[\widehat{Q}_c^{\text{UCB}}(s,a)\Big]\Big)\Big\}$

13:        $\psi_i \leftarrow \psi_i - \alpha_Q \mathbb{E}_{s,a\sim\mathcal{B}}\Big[\nabla_{\psi_i}\Big(c(s,a) + \gamma\mathbb{E}_{a'\sim\pi_\phi(\cdot|s')}\widehat{Q}_c^{\psi_i^-}(s',a') - \widehat{Q}_c^{\psi_i}(s,a)\Big)^2\Big], i = 1,\dots,E$

14:        $\theta_i \leftarrow \theta_i - \alpha_Q \mathbb{E}_{s,a\sim\mathcal{B}}\Big[\nabla_{\theta_i}\Big(r(s,a) + \gamma\min_i \mathbb{E}_{a'\sim\pi_\phi(\cdot|s')}\widehat{Q}_r^{\theta_i^-}(s',a') - \widehat{Q}_r^{\theta_i}(s,a)\Big)^2\Big], i = 1,2$

15:        $\lambda \leftarrow \max\Big\{0, \lambda - \alpha_\lambda \mathbb{E}_{s,a\sim\mathcal{B}}\Big[d - \widehat{Q}_c^{\text{UCB}}(s,a)\Big]\Big\}$

16:        $\psi_i^- \leftarrow \tau\psi_i + (1-\tau)\psi_i^-, i = 1,\dots,E$

17:        $\theta_i^- \leftarrow \tau\theta_i + (1-\tau)\theta_i^-, i = 1,2$

18:     **end for**

19: **end for**

---

# D   EXPERIMENT DETAILS

## D.1   ENVIRONMENT DESCRIPTIONS

**Safety-Gym** In Safety-Gym (Ray et al., 2019), the agent perceives the world through its sensors and interacts with the world through its actuators. In **Button** tasks, the agent needs to reach the goal while avoiding both static and dynamic obstacles. In **Push** tasks, the agent needs to move a box to a series of goal positions. Dense positive rewards will be given to the agent for moving towards the goal, and a sparse reward will be given for successfully reaching the goal. The agent will be penalized through the form of a cost for violating safety constraints, i.e., colliding with the static obstacles or pressing the wrong button. To accelerate training, we adopt the modifications on Safety-Gym tasks made by (Liu et al., 2022). The comparison is still valid since all the algorithms are evaluated on the same set of tasks.

**Velocity-Constrained MuJoCo** In MuJoCo tasks (Todorov et al., 2012), we follow the cost definition and constraint threshold used in (Zhang et al., 2020; Yang et al., 2022) for reproducible comparison. Specifically, the cost is calculated as the velocity of the agent, and the threshold of each task is set to $50\%$ of the velocity attained by an unconstrained PPO (Schulman et al., 2017) agent after training for a million samples.

**oCPM** At the beginning of each day, advertiser $s$ assigns the target ROI $Roi_s$ and the maximum total pay $P_s$. At time step $t$, the advertising system bids $pay_{s,t}$ for advertiser $s$. If this bid succeeds, then the advertiser expends $pay_{s,t}$ and obtains revenue $gmv_{s,t}$ (the gross merchandise value) from the exhibited advertisements. In this process, the system has to ensure that the average conversion

is larger than the target $Roi_s$, i.e.,

$$\frac{\sum_t gmv_{s,t}}{P_s} > Roi_s \iff \sum_t -gmv_{s,t} < -Roi_s \cdot P_s, \tag{28}$$

Thus, the optimization problem in this scenario can be formulated as:

$$\max \sum_t pay_{s,t}$$
$$\text{s.t. } \sum_t -gmv_{s,t} < -Roi_s \cdot P_s, \forall s \tag{29}$$

We further formulate this problem as a constrained Markov Decision Process:

- The state consists of the user context features, ads context features, cumulative revenue, cumulative pay, maximum total pay, target ROI, $pCTR$ (predicted click-through rate), $pCVR$ (predicted conversion rate) and $pGMV$ (predicted gross merchandise volume) of one advertiser.
- The action is to output a weight $k \in [0.5, 1.5]$, which controls the bidding by formula $bid = k \cdot \frac{pCTR \cdot pCVR \cdot pGMV}{Roi_s}$.
- The reward is $pay_{s,t}$ for a successful bid and $0$ otherwise.
- The cost is $-gmv_{s,t}$ for a successful bid and $0$ otherwise.
- The constraint budget is $-Roi_s$. Note that by definition the ROI value is computed by $\frac{\sum_t pay_{s,t}}{\sum_t gmv_{s,t}}$, thus in this scenario we replace $\mathbb{E}[Q_c^\pi] - d$ in the algorithm by $\mathbb{E}[\frac{Q_r^\pi}{Q_c^\pi}] + Roi_s$. Furthermore, the frequency of the change in the target ROI is relatively low and does not affect the training.

In this task, the cost and the constraint budget are negative, which means that the constraint is violated at the starting stage of bidding and can only be satisfied over time. During evaluation, each advertiser switches among policies trained by different algorithms, in order to eliminate the differences between advertisers and make the online experiment fairer.

## D.2 COMPARISONS WITH ON-POLICY BASELINES

The on-policy baselines include 1) PDO (Chow et al., 2017), which applies Lagrangian relaxation to devise risk-constrained RL algorithm, 2) RCPO (Tessler et al., 2018), which incorporates the constraint as a penalty into the reward function to guide the policy towards a constraint-satisfying one, 3) IPO (Liu et al., 2020), which augments the objective with a logarithmic barrier function as a penalty, 4) CRPO (Xu et al., 2021), which updates the policy alternatingly between reward maximization and constraint satisfaction, and 5) CUP (Yang et al., 2022), which applies a conservative policy projection after policy improvement.

Figure 6 shows the comparative evaluation on two benchmark task sets. As shown by the x-axes, the total number of samples used by on-policy methods are set to 5 million in order to present the asymptotic performance of these methods. Note that the curves of CAL look squeezed in these plots. This is because the x-axes are altered to a much larger scale compared to those used in Figure 4. We can see that in each task, CAL achieves a reward performance comparable to the on-policy baseline that yields the highest asymptotic reward in this task. Although IPO attains higher reward in several Safety-Gym tasks, its performance on MuJoCo tasks is apparently inferior to CAL. Moreover, it is worth noting that no on-policy baseline is able to achieve constraint satisfaction on all these tested tasks. Most importantly, CAL's advantage in sample efficiency is significant, and this is particularly important in safety-critical applications where interacting with the environment could be potentially risky.

## D.3 IMPLEMENTATION DETAILS & HYPERPARAMETER SETTINGS

**Baselines** We use the implementations in the official codebases, i.e.,

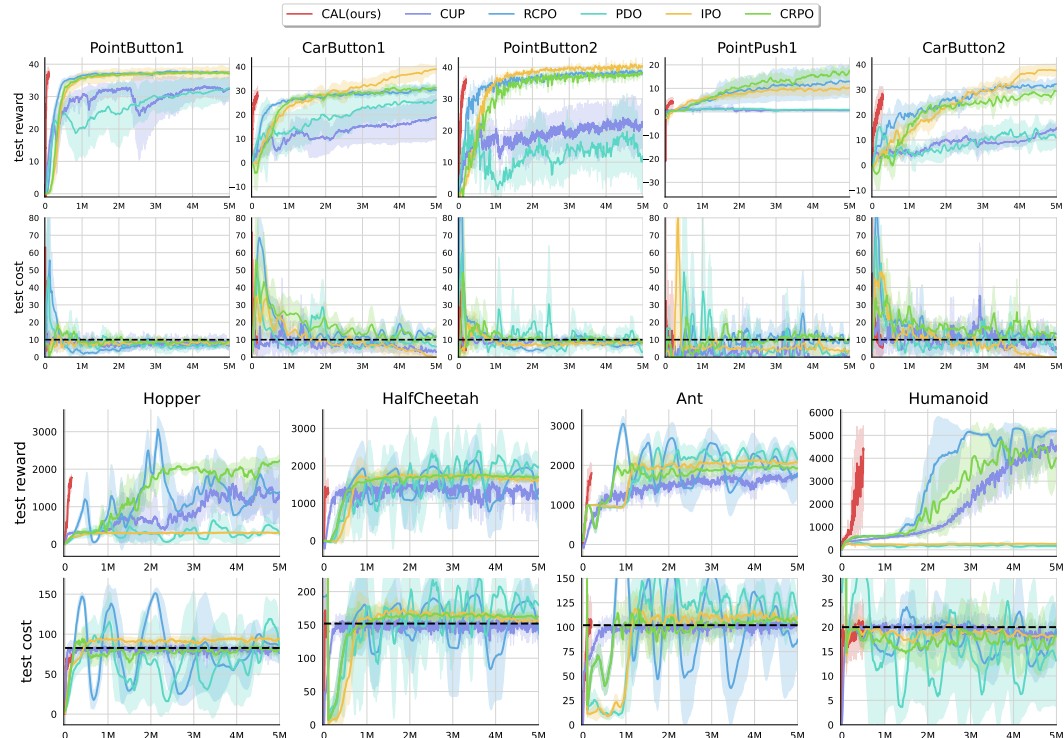

Figure 6: Comparisons with on-policy baselines on Safety-Gym (top half) and velocity-constrained MuJoCo (bottom half).

- https://github.com/AlgTUDelft/WCSAC for WCSAC (Yang et al., 2021)
- https://github.com/liuzuxin/cvpo-safe-rl for CVPO (Liu et al., 2022)
- https://github.com/zmsn-2077/CUP-safe-rl for CUP (Yang et al., 2022)
- https://github.com/PKU-Alignment/omnisafe for PDO (Chow et al., 2017), RCPO (Tessler et al., 2018), IPO (Liu et al., 2020), CRPO (Xu et al., 2021). Note that this is an infrastructural framework for safe RL algorithms implemented by Ji et al. (2023).

For SAC-PIDLag, we use the implementation in the CVPO codebase. We adopt the default hyperparameter settings in the original implementations. Note that the MuJoCo results are not presented in the CVPO paper, so the hyperparameters of CVPO are finetuned on MuJoCo tasks based on the authors' modified codebase [7] (https://github.com/liuzuxin/fsrl). In addition, the risk hyperparameter $\alpha$ in WCSAC is set to the risk-averse 0.1.

**CAL** The network structures of the actor and the reward/cost critics of CAL are all the same with the off-policy baselines, i.e., 256 neurons for two hidden layers with ReLU activation. The discount factor $\gamma$ is set to 0.99 both in reward and cost value estimations. The optimization of the networks is conducted by Adam with the learning rate $5e - 4$ for the cost critics, and $3e - 4$ for the actor and the reward critics. The ensemble size $E$ is set to 4 for MuJoCo tasks and 6 for Safety-Gym tasks. The conservatism parameter $k$ is set to 0.5 for all tasks except for PointPush1 (0.8). The convexity parameter $c$ is set to 10 for all tasks except for Ant (100), HalfCheetah (1000) and Humanoid (1000).

**About the Up-To-Data Ratio** To make a more sufficient utilization of each data point and improve the sample efficiency, a larger up-to-data (UTD) ratio (i.e., the number of updates taken by an agent per environment interaction) (Chen et al., 2021) than normal off-policy algorithms is adopted in the implementation of CAL. The UTD ratio of CAL is set to 20 for all tasks except for Humanoid

---

[7]Specifically, we set 1) the number of episodes per data collection to 1; 2) the number of updates per environment sample to 0.5; and 3) the constraint of the policy's standard deviation in the M-step to 0.005.

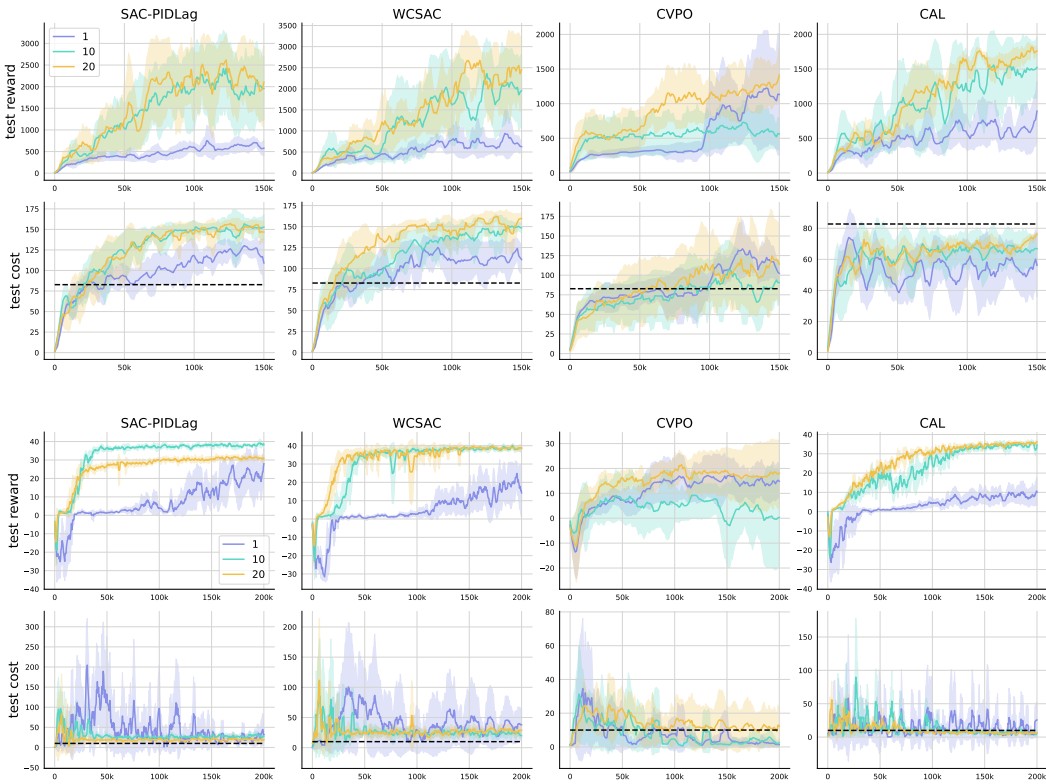

Figure 7: Results of different UTD settings (1, 10 and 20) on Hopper (top half) and PointButton2 (bottom half).

(10) and HalfCheetah (40). In the comparative evaluation (Section 4.1), we adopt the default UTD settings for the baselines [8], which are lower than the ratio in CAL. Since the off-policy baselines also have the potential to be improved by using a larger UTD ratio, here we provide results of different UTD settings for these baselines (and also CAL) for a more comprehensive comparison. Clearly, we can see from Figure 7 that different from CAL, increasing the UTD ratios of the baselines only improves their reward performances but cannot improve their constraint satisfactions, indicating that the capability of using a large UTD ratio is also one of the non-trivial advantages of CAL.

### D.4 ABLATION STUDY FOR THE ENSEMBLE SIZE

See Figure 8.

## E RELATED WORK

In recent years, deep safe RL methods targeting at high-dimensional controlling tasks have been well developed. One of the most commonly adopted approach for safe RL is the primal-dual framework due to its simplicity and effectiveness. In this regard, Chow et al. (2017) propose a primal-dual method and prove the convergence to constraint-satisfying policies. Tessler et al. (2018) propose a multi-timescale approach and use an alternative penalty signal to learn constrained policies with local convergence guarantees. Paternain et al. (2019) prove that primal-dual methods have zero duality gap under certain assumptions. Ding et al. (2020) show the convergence rate of a natural policy gradient based on the primal-dual framework. Ray et al. (2019) make practical contributions by benchmarking safe RL baselines on the Safety-Gym environment. Nevertheless, the majority of existing primal-dual methods are on-policy by their design. Despite the convergence guarantees,

---

[8]except for SAC-PIDLag in PointButton2 (20) and CVPO in MuJoCo tasks (0.5).

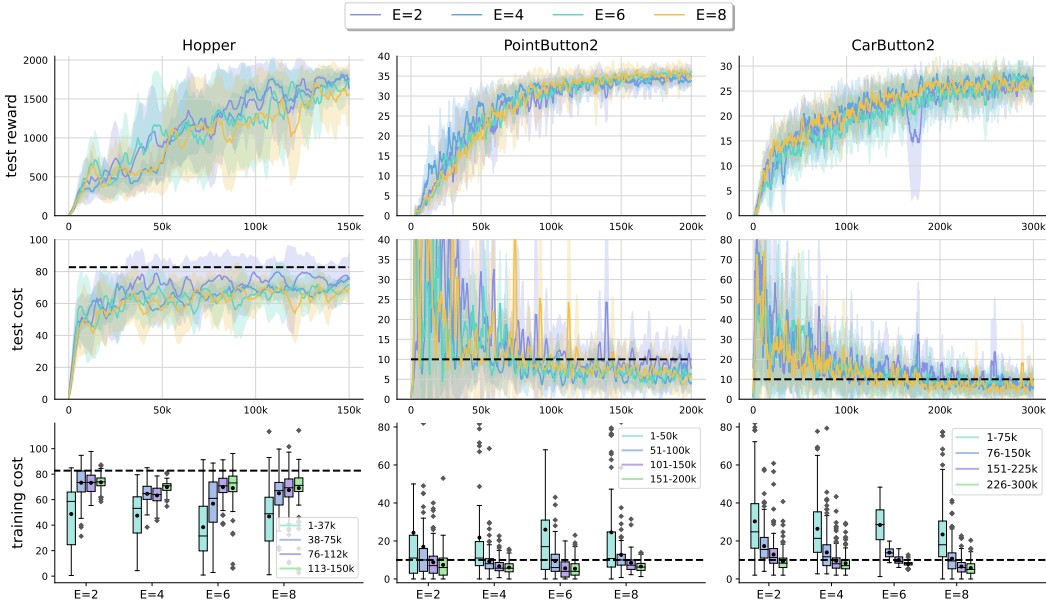

Figure 8: Ablation study for the ensemble size $E$.

these methods usually encounter the sample efficiency issues, demanding a large amount of environment interaction and also constraint violations during the training process. This may impede the deployment of such methods in safety-critical applications. An approach to avoid risky environment interaction is training constraint-satisfying policies using only offline data (Le et al., 2019; Lee et al., 2022a; Xu et al., 2022a; Liu et al., 2023; Lin et al., 2023). Instead, this paper focuses on the online setting and attempts to enhance the safety of RL methods by improving the sample efficiency.

The methods most related to ours are as follows: Stooke et al. (2020) apply PID control on the dual variable and show its promising performance when combined with the on-policy PPO method (Schulman et al., 2017). This method is adopted as one of our baselines since as reported in (Stooke et al., 2020), the PID control can stabilize the update of the dual variable and thus has the potential to be adapted to the off-policy setting. WCSAC (Yang et al., 2021) augments the naïve SAC-Lag method with a variance estimator of the cost value to achieve risk control. Yet, in practice we found that the temporal-difference-style learning of this variance estimator is less stable than directly using the variance of an ensemble, which may be one of the causes explaining WCSAC's inferior performance compared to CAL. CVPO (Liu et al., 2022) allows off-policy updates by breaking the brittle bond between the primal and the dual update and introducing a variational policy to bridge the two updates. This method shows significant improvement in sample efficiency but suffers from relatively low computational efficiency, due to the frequent sampling operations in the action space when computing the variational policy.

