# OpenReview forum: "Off-Policy Primal-Dual Safe Reinforcement Learning"
_ICLR.cc/2024/Conference — ICLR 2024 poster_

### Official Review · Reviewer_oKob · 2023-10-28

**Soundness:** 2 fair
**Presentation:** 3 good
**Contribution:** 2 fair
**Rating:** 5
**Confidence:** 4

**Summary:**

The study conducted herein delves into the issue of underestimation in safe off-policy RL. In addressing this problem, the study conducts a series of experiments aimed at assessing the efficacy of the proposed methodology. The outcomes of these experiments hold substantial promise for enhancing RL applications within real-world environments.

**Strengths:**

1. The presentation is clear and straightforward, making it easy to understand.
2. The research conducted in this study is interesting.

**Weaknesses:**

1. The study could benefit from a more comprehensive exploration of convergence and stability, particularly concerning policy optimization. The theoretical analysis in this regard appears to be somewhat limited.

2. It is suggested that the on-policy experiments be completed, and the results of training for on-policy RL methods be included for a more comprehensive evaluation. The on-policy experiments for this method are not completed.

**Questions:**

1. Is it possible to extend Equation (5) into a third-order equation, and if so, what impact might this have on performance?
2. What is the connection between the variance in Q-cost value estimation and the process of policy optimization?
3. How does the value of "c" affect performance in terms of both cost minimization and reward maximization?

---

> ### Author Response · Authors · 2023-11-16
> **Responses to weaknesses**
>
> We thank the reviewer for all the valuable comments. Please refer to the point-to-point responses below.
>
> ##  W1: About the theoretical analysis in terms of convergence and stability in policy optimization.
>
> We agree that this work can benefit from a thorough theoretical analysis in terms of convergence and stability. However, we have to clarify that the main contribution of this work is in the extensive experimental campaign, as also argued by reviewer cCE3. To our best knowledge, CAL is the first off-policy primal-dual method that achieves asymptotic performance comparable to state-of-the-art on-policy methods in such a low data regime. The theoretical part in this paper proves the invariance of the local optima under the ALM modification, and more importantly, explains the stabilization effect of ALM from an energy perspective and thus provides an insight into how the two proposed ingredients interplay and how this benefits primal-dual policy optimization, which are then well-verified by our interpretative experiments (e.g., Figure 3). We believe that all these endeavors can be notable contributions to the safe RL community.
>
>
>
> ##  W2: About the on-policy experiments.
>
> Thanks for the suggestion. To clarify, the reasons why we trained CAL using fewer samples than that for on-policy baselines is that the performance of CAL can be seen as near convergence in the low data regime presented in Figure 4. However, directly transferring these curves to Figure 6 may make them look incomplete because the x-axes are altered to a much larger scale since on-policy baselines requires much more samples to converge. Also, it is worth noting that the comparative evaluation against on-policy baselines is sufficient to demonstrate CAL's effectiveness.
>
> Still, we agree that it would be better to present CAL's performance in a higher data regime. Thus, we are currently trying to run CAL as many simulation steps as possible in order to obtain a true asymptotic performance, and we will append this result to our future revision.

---

> ### Author Response · Authors · 2023-11-16
> **Responses to questions**
>
> ##  Q1: Is it possible to extend Equation (5) into a third-order equation, and if so, what impact might this have on performance?
>
> The goal of adding the quadratic penalty term to Equation (5) is to convexify the neighborhood area of a locally optimal solution. More specifically, because the second-order derivative of this term is positive definite, the incorporation of quadratic penalty with a large penalty weight can ensure that the augmented Lagrangian has a positive definite second-order derivative in the vicinity of local optima, thereby promoting the convergence of our algorithm (See Appendix B). In comparison, employing the third-order penalty cannot achieve this goal because its second derivative is not guaranteed to be positive definite.
>
> Still, one might consider using a penalty of a higher order, such as fourth, sixth, etc. However, incorporating such a penalty leads to more hyperparameter tuning, and our experiment results show that we can already obtain a satisfactory performance by tuning the hyperparameter $c$.
>
>
>
> ##  Q2: What is the connection between the variance in Q-cost value estimation and the process of policy optimization?
>
> In the expression of UCB (i.e., Eq. (3)), the variance of the $Q\_c$ ensemble serves as an indicator of epistemic uncertainty and ensures that the $\\widehat{Q}\_c^{\\text{UCB}}$ can upper bound the cost value of most state-action pairs. Therefore, it induces conservativeness by shrinking the policy search space (i.e., $\\mathbb{E}[Q^\\pi\_c]\\leq \\mathbb{E}[\\widehat{Q}^{\\text{UCB}}\_c]$ implies $\\{\\pi |\\mathbb{E}[\\widehat{Q}^{\\text{UCB}}\_c]\\leq d \\}\\subseteq \\{\\pi |\\mathbb{E}[Q^\\pi\_c]\\leq d \\}$), as indicated by the conservative boundary in Figure 1. More specifically, by adding the standard deviation in cost estimation, the policy can be optimized to favor actions that not only result in high reward and constraint satisfaction but also yield low uncertainty in cost estimation. This is because to lower the value of $\\widehat{Q}\_c^{\\text{UCB}}$, the policy have to choose actions that not only have low safety costs estimated by each $\\widehat{Q}\_c^i$ but also lead to low variance among the estimations of different $\\widehat{Q}\_c^i$.
>
>
>
> ##  Q3: How does the value of "c" affect performance in terms of both cost minimization and reward maximization?
>
> In theory, the value setting of $c$ controls the degree of convexity and also stability in the neighborhood of local optima. Increasing this value can help stabilizing policy learning, but an overly large $c$ imposes a large penalty on policies exceeding the conservative boundary (where $\\mathbb{E}[\\widehat{Q}^{\\text{UCB}}\_c]>d$), which significantly slows down the process of expanding the conservative boundary as shown in Figure 1. According to the ablation studies in Section 4.2, a larger $c$ generally leads to fewer performance oscillations in terms of both cost and reward, but an overly large $c$ can also hinder reward maximization by having the policy stuck in highly conservative solutions where both the reward and the cost are low.

---

### Official Review · Reviewer_M5JF · 2023-11-01

**Soundness:** 1 poor
**Presentation:** 3 good
**Contribution:** 2 fair
**Rating:** 6
**Confidence:** 4

**Summary:**

This paper presents primal-dual method to solve constrained RL problem combining the conservative policy optimization and local convex optimization. By conservative policy, the estimated Q-value is calculated via averaging over several copies of Q-value and adds UCB-style bonus term. The local convexification comes from well-posed augmented Lagrangian problem.

**Strengths:**

The primal-dual approach is known to be difficult to train but combining several technique, the authors show improved performance compared to recent results in safe RL.

**Weaknesses:**

- I believe there needs to be some clarifications about the proof that ALM does not modify or alter the position of local optima. Regarding equation (20), in general expectation of max or square is not same as the max or square of expectation, i.e., $\mathbb{E}[X]^2\leq \mathbb{E}[X^2]$ and the same for the max operator. Hence, we cannot guarantee the following equation to hold:
$$ |\max\\{ 0,\lambda+c \mathbb{E} [ \hat{Q}^{UCB}_c(s,a) ] \\}  |^2 = \mathbb{E} [| \max\\{ 0,\lambda+c\hat{Q}^{UCB}_c(s,a) \\} |^2].$$

**Questions:**

- Why do we have to reformulate the augmented Lagrangian? What is the advantage of the reformulation according to Luenberger compared to the original augmented Lagrangian (equation (13))?

- For Figure (5), as for Hopper environment, when $c=0$, the test-cost seems to be not that bad in the sense that average is below the threshold. Does this imply the threshold constraint is too loose?

- Please provide page number or section for Luenberger 1984, about the derivation of augmented Lagrangian, which will be helpful to the readers.


- In section 3.1, about the underestimation, we have $\mathbb{E} [ \hat{Q}_c(s,\pi(s))]=\mathbb{E}[Q_c(s,\pi(s))+\epsilon]=\mathbb{E} [Q_c(s,\pi(s))]$. Why does taking minimum over $\pi$ implies underestimation bias? Please give some clarifications.

- It would be helpful if the authors formally introduce and rephrase the theorem in A.M. Duguid.


A.M. Duguid. Studies in linear and non-linear programming, by k. j. arrow, l. hurwicz and h. uzawa. stanford university press, 1958. 229 pages. 7.50. Canadian Mathematical Bulletin, 3(3):196–198, 1960. doi: 10.1017/S0008439500025522.

---

> ### Author Response · Authors · 2023-11-16
>
> We thank the reviewer for all the valuable comments. Please refer to the point-to-point responses below.
>
> ##  W: Clarification about the proof that ALM does not alter the position of local optima.
>
> Thank you very much for pointing out this issue, and we clarify that this actually results from a typing error.
>
> Specifically, in our original submission, we mistyped the actual objective
>
> \begin{equation}
> \min\_{\lambda\geq 0}\max\_{\pi}\mathbb{E}\_{s\sim \rho\_\pi,a\sim\pi(\cdot|s)}\left[\widehat{Q}\_r^\pi(s,a)\right] - \frac{1}{2c}\left[\left( \max \{0, \lambda-c(d-\mathbb{E}\_{s\sim \rho\_\pi,a\sim\pi(\cdot|s)}\left[\widehat{Q}\_c^{\text{UCB}}(s,a)\right]) \} \right)^2-\lambda^2\right]
> \end{equation}
>
> as
> \\begin{equation}
> \\min\_{\lambda\geq 0}\\max\_{\\pi}\\mathbb{E}\_{s\\sim \\rho\_\pi,a\\sim\\pi(\\cdot|s)}\\left\\{\\widehat{Q}\_r^\\pi(s,a) - \\frac{1}{2c}\\left[\\left( \\max \\{0, \\lambda-c(d-\\widehat{Q}\_c^{\\text{UCB}}(s,a) \\} \\right)^2-\\lambda^2\\right] \\right\\}
> \\end{equation}
> by missing an inner expectation.
>
> Apparently, the actual objective does not require an interchange between the expectation and the max or square operator, and thus does not cause the trouble mentioned by the reviewer in the proof that ALM does not alter the position of local optima.
>
> To see why we actually used the objective with an inner expectation, we can first transform this objective into an equivalent form (i.e., Objective (4a)(4b) in our revision, page 4):
>
> \\begin{cases}
>     \\min\\limits\_{\\lambda\\geq 0}\\max\\limits_{\\pi}\\mathbb{E}\\left[\\widehat{Q}\_r^\\pi\\right] - \\lambda\\left(\\mathbb{E}\\left[\\widehat{Q}\_c^{\\text{UCB}}\\right] - d\\right)-\\frac{c}{2}\\left(\\mathbb{E}\\left[\\widehat{Q}\_c^{\\text{UCB}}\\right] - d\\right)^2,&\\text{if}\ \\frac{\\lambda}{c}>d-\\mathbb{E}[\\widehat{Q}\_c^{\\text{UCB}}] \\\\
>     \\min\\limits\_{\\lambda\\geq 0}\\max\\limits\_{\\pi}\\mathbb{E}\\left[\\widehat{Q}\_r^\\pi\\right] + \\frac{\\lambda^2}{2c} ,&\\text{otherwise}.
> \\end{cases}
>
> When $\frac{\lambda}{c}\leq d-\mathbb{E}[\widehat{Q}\_c^{\text{UCB}}]$, the objective does not involve $\widehat{Q}\_c^{\text{UCB}}$, which is reasonable since we have $d\geq\mathbb{E}[\widehat{Q}\_c^{\text{UCB}}]$. When $\frac{\lambda}{c}>d-\mathbb{E}[\widehat{Q}\_c^{\text{UCB}}]$, differentiating the objective w.r.t. the policy yields
>
> \\begin{equation}
> \\mathbb{E}\\left[\\nabla\_\\pi\\widehat{Q}_r^\\pi\\right]-(\\lambda-c(d-\\mathbb{E}\\left[\\widehat{Q}\_c^{\\text{UCB}}\\right])\\cdot\\mathbb{E}\\left[\\nabla\_\\pi\\widehat{Q}\_c^{\\text{UCB}} \\right],
> \\end{equation}
> which is equivalent to
>
> \begin{equation}
> \mathbb{E}\left[\nabla\_\pi\widehat{Q}\_r^\pi-(\lambda-c(d-\mathbb{E}\left[\widehat{Q}\_c^{\text{UCB}}\right])\cdot\nabla\_\pi\widehat{Q}\_c^{\text{UCB}} \right].
> \end{equation}
>
> By replacing $\nabla\_\pi\widehat{Q}\_r^\pi$ with the SAC objective, this gradient corresponds exactly to the policy update in our pseudo code (Line 11 of Algorithm 1 in the **original version** of our paper) and also to the actual implementation (Lines 163-208 in `./CALsubmission/sac/cal.py` in `CAL_code.zip` which can be found in our supplementary materials).
>
> As for the dual update for $\lambda$, according to this objective, we should perform $\\lambda \\leftarrow \\max\\left\\{0, \\lambda - \\alpha\_\\lambda \\mathbb{E}\_{s,a\\sim\\mathcal{B}} \\left[d-\\widehat{Q}\_c^{\\text{UCB}}(s,a) \\right]\\right\\} $ when $\frac{\lambda}{c}> d-\mathbb{E}[\widehat{Q}\_c^{\text{UCB}}]$, and perform $\lambda\leftarrow \max\\{0,\lambda-\frac{\lambda}{c}\\}$ otherwise. However, empirical results show that exclusively performing the former update rule is sufficient to obtain a satisfactory performance, so we adopt this form in our pseudo code (Line 14 of Algorithm 1 in the original version of our paper) and also our implementation (Lines 221-229 in `./CALsubmission/sac/cal.py`).
>
> We have modified corresponding parts in our paper (Section 3.2 and Appendix A). Please refer to our revision and let us know if there is still any confusion.

---

> ### Author Response · Authors · 2023-11-16
>
> ##  Q1: Why do we have to reformulate the augmented Lagrangian?
>
> According to our above clarification about the proof that ALM does not alter the position of local optima, the two formulations are equivalent after we add the missing inner expectation in the reformulated objective. To make the objective more intuitive and easier to understand, in the revised version, we have replaced the objective in Section 3.2 with:
> \\begin{equation}
> \\begin{cases}
>     \\min\\limits\_{\\lambda\\geq 0}\\max\\limits\_{\\pi}\\mathbb{E}\\left[\\widehat{Q}\_r^\\pi\\right] - \\lambda\left(\\mathbb{E}\\left[\\widehat{Q}\_c^{\\text{UCB}}\\right] - d\\right)-\\frac{c}{2}\\left(\\mathbb{E}\\left[\\widehat{Q}\_c^{\\text{UCB}}\\right] - d\\right)^2,&\\text{if}\\ \\frac{\lambda}{c}>d-\\mathbb{E}[\\widehat{Q}\_c^{\\text{UCB}}] \\\\
>     \\min\\limits\_{\\lambda\\geq 0}\\max\\limits\_{\\pi}\\mathbb{E}\\left[\\widehat{Q}\_r^\\pi\\right] + \\frac{\\lambda^2}{2c} ,&\\text{otherwise}.
> \\end{cases}
> \\end{equation}
> The reason why we did not use this form in our original manuscript is that we wanted to simplify the objective into one single line. However, as the reviewer suggests, this might cause confusion, so we have modified our manuscript to mitigate this issue.
>
>
>
> ##  Q2: About the constraint threshold in Hopper.
>
> In theory, according to our description of how ALM helps addressing the instability and over-conservativeness issue, $c=0$ (without ALM) means that the performance can become instable and over-conservative. Therefore, the fact that **"when $c=0$, the test-cost seems to be not that bad in the sense that average is below the threshold"** in Hopper does not contradict the above statement since the reward in this case oscillates violently and is overly conservative.
>
>
>
> ##  Q3: About the page number for [Luenberger et al., 1984].
>
> Thanks for the suggestion. The part relevant to the derivation of augmented Lagrangian in [Luenberger et al., 1984] is Pages 445-453. We have added the page number to the place where the ALM objective first appears (Section 3.2) in our revision.
>
>
>
> ##  Q4: Why does taking minimum over $\\pi$ implies underestimation bias?
>
> We misplaced the min operator in the inequality in Section 3.1, and the true inequality we used is actually$ \\mathbb{E}\_\\epsilon [{\\min_\pi}({Q}\_c(s,\\pi(s)) +\\epsilon)]\\leq\\min\_\\pi {Q}\_c(s,\\pi(s)) $ for any given $s$. We have corrected this in the revision.
>
> In terms of why this inequality is generally true, there is an intuitive experiment in [Thrun
> & Schwartz, 1993] (Section 2) that helps illustrate this phenomenon, which is also used to serve as the motivation of addressing the overestimation bias in $Q$ function approximation in the TD3 paper (Section 4 in [Fujimoto et al., 2018]). The key point here is that due to the function approximation noise, some of the Q-values might be too small, while others might be too large. The min operator, however, always picks the smallest value, making it particularly sensitive to underestimations.
>
>
>
> ##  Q5: Introduce and rephrase the theorem in A.M. Duguid.
>
> Thanks for the advice. We have introduced the corresponding theorem as a lemma in Appendix B, where we also show how this lemma leads to the fact that there exists a $c^*>0$ such that if $c>c^*$, the damping matrix $A$ is positive definite at the constrained minima.

---

> > ### Comment · Reviewer_M5JF · 2023-11-17
> > **Response to authors**
> >
> > Thank you for the detailed response. Most of my concerns have been addressed.
> >
> > It would be better to have a numbering in the proposition in the revised manuscript, e.g., Proposition 1.
> >
> > Overall I have increased my score to 6.

---

> > > ### Author Response · Authors · 2023-11-17
> > >
> > > Thanks for the reply and the suggestion. We have updated our revision to add the number to the Proposition in the main paper.

---

### Official Review · Reviewer_Dfxp · 2023-11-03

**Soundness:** 4 excellent
**Presentation:** 4 excellent
**Contribution:** 3 good
**Rating:** 8
**Confidence:** 3

**Summary:**

This paper considers the problem of constrained off-policy reinforcement learning and remedies the issue of inaccurate cost estimation in prior Lagrangian duality-based approaches in two steps: i) the cost value estimate is replaced by an upper-confidence bound, which is derived through the uncertainty, or lack of consensus, in an ensemble of cost value estimators, and ii) the regular Lagrangian is replaced by the augmented version of the Lagrangian in order to make the optimization landscape in the feasible region surrounding the local optimal policy. Essentially, the first component encourages the policy to be conservative and ensures better constraint satisfaction, while the second component encourages the policy to become as aggressive as possible in the space of feasible policies in terms of the primary reward function. Numerous experiments demonstrate the superiority of the proposed method, termed CAL, over multiple baseline methods in a variety of settings and environments.

**Strengths:**

- Training safe RL policies is a very important research direction, and alongside the aforementioned components of controlled conservativism and aggressiveness, the paper presents a sample-efficient constrained RL training procedure that also has safer behavior during **training**, something that has potentially been overlooked in prior work in this area.

- I very much enjoyed reading the manuscript. The content is very well presented, and the experimental results are, in my opinion, excellent and convincing as to why and how the proposed method works well.

**Weaknesses:**

One limitation I can think of is the inclusion of only one constraint in the problem formulation, solution, and experiments. The authors have mentioned that having a single constraint does not prevent generality, but I wonder how the method performs in settings with multiple competing constraints, especially as compared to baseline methods. A discussion on how the two introduced components interplay in the presence of several constraints would be helpful.

**Questions:**

- Could you please provide more details on how the cost value ensemble in (3) is created? Do you have multiple (i.e., $E$) parallel parameterized models that each estimate the cost in parallel? Should the expectation in (3) be replaced by an empirical mean (i.e., $\frac{1}{E} \sum_{i=1}^E$)?
- The last statement in Section 3.1 mentions that the UCB "encourages the policy to choose actions with low uncertainty in cost estimation." Could you comment on why that is the case? Based on (3), it seems that for state-action pairs with low uncertainty in cost estimation, the UCB is close to each of the ensemble estimates, but it does necessarily translate to the action leading to the highest estimated reward in that state.
- I suggest changing the notation for the hyperparameter $c$ in the augmented Lagrangian, as it has also been used to denote the cost function in the CMDP definition.

---

> ### Author Response · Authors · 2023-11-16
>
> We thank the reviewer for the valuable comments. Please refer to the point-to-point responses below.
>
> ##  W: About the performance with multiple competing constraints.
>
> We agree that this is an interesting research problem. Theoretically, there is no essential difference when switching to the multi-constraint setting since we can simply replace the scalar $Q\_c$ functions and the dual variable with a vector. However, in practice, depending on how these constraints relate to each other, the difficulty in balancing different constraints in policy optimization may vary.
>
> One practically meaningful constraint we can come up with in MuJoCo tasks is the amount of torque of the agent's joints (a large torque may shorten the lifespan of the motor). Yet, we are not sure if this torque constraint would compete with the speed constraint used in our paper, and we are currently investigating this. To be more specific, we are conducting experiments to determine an appropriate value for this constraint in different tasks and for different joints in a single robot. Also, we are trying to incorporate this constraint with the speed constraint to see how the methods perform in such a setting. We will append these results in our future revision as soon as we have finished this experiment.
>
>
>
> ##  Q1: About the cost value ensemble.
>
> > How the cost value ensemble in (3) is created?
>
> Specifically, we maintain $E$ bootstrapped $Q\_c$ networks, i.e., $\\{\\widehat{Q}^i\_c \\}\_{i=1}^E$, which share the same architecture and only differ by the initial weights. These weights are initialized by sampling from a normal distribution (The corresponding part in our code: `torch.nn.init.xavier_uniform_(m.weight, gain=1)`). All the networks are trained on the same dataset.
>
> > Do you have multiple (i.e., $E$) parallel parameterized models that each estimate the cost in parallel?
>
> Yes. During cost estimation, each $\\widehat{Q}^i\_c$ estimates the cumulative cost in parallel and independently.
>
> > Should the expectation in (3) be replaced by an empirical mean (i.e., $\\frac{1}{E}\\sum\_{i=1}^{E}$)?
>
> Yes, the reviewer is right. It would be more accurate to use an empirical mean, and we have modified this definition in our revision.
>
>
>
>
>
> ##  Q2: About the statement that UCB "encourages the policy to choose actions with low uncertainty in cost estimation."
>
> Yes, the reviewer is right. Actions favored by the conservative policy optimization are those not only yield low uncertainty in cost estimation but also result in high reward and constraint satisfaction. We have modified the corresponding expression in our revision (Section 3.1).
>
>
>
> ##  Q3: About the notation for the hyperparameter $c$ in ALM.
>
> Thanks for the suggestion. We will change the notation to avoid confusion in our future revision, but we have to temporarily maintain this notation in the current revision (available in this webpage) since some questions from other reviewers are related to this notation.

---

> > ### Comment · Reviewer_Dfxp · 2023-11-20
> > **Thank you!**
> >
> > Thank you very much for your response. I will keep my score unchanged and maintain a favorable view of the paper. It would be great if additional MuJoCo experiments with multiple constraints could be added, which would further strengthen the paper.

---

### Official Review · Reviewer_cCE3 · 2023-11-04

**Soundness:** 3 good
**Presentation:** 3 good
**Contribution:** 3 good
**Rating:** 8
**Confidence:** 4

**Summary:**

This paper considers the problem of safe reinforcement learning in constrained Markov decision processes, where the agent tries to maximize reward while keeping a separate cost signal under a given threshold. The authors propose an off-policy primal-dual deep RL method that combines two techniques. The first, called conservative policy optimization, replaces cost-value estimates with an empirical upper confidence bound based on the variance of the estimates across an ensemble of Q networks. The second, called local policy convexification, applies the augmented Lagrangian method to the primal-dual formulation of constrained RL. While the first is intended to make the policy search more conservative reducing the number of constraint violations due to cost estimation, the second is intended to stabilize the learning process. The algorithm that combines these two techniques, called CAL (Conservative Augmented Lagrangian), is tested on benchmark tasks and on a real advertising system (in a semi-batch fashion) and compared with several baselines, showing a significant improvement over the state of the art both in terms of reward maximization and constraint violation.

**Strengths:**

The paper considers an important problem, that of safe RL, from a practical perspective.
Its main strength is in the extensive experimental campaign. The experiments are well designed and documented, and convincingly show the advantages of the proposed method over the state of the art. It is particularly impressive to see experiments on a real system, where the proposed method is also compared against several baselines.
The main point of originality is in using the augmented Lagrangian method in this setting.

**Weaknesses:**

The theoretical motivations are a bit brittle. The authors should be more clear on where the theoretical inspiration ends and where heuristics begin. For instance, in section 3.1, it is suggested that Equation (3) would be an actual upper confidence bound in a linear MDP, which is inaccurate, as the upper confidence bound for linear MDPs has a specific form that does not correspond to equation (3).
Also, more words should be spent in explaining how the bootstrapped value ensemble is constructed, and why you chose this form of uncertainty estimation.
Regarding ALM, I think that the theory that is provided in the appendix is interesting and deserves more space in the main paper.

**Questions:**

How many Q networks did you use in the ensemble and how did you choose this hyperparameter?

Just a minor remark: the objective should be stated in terms of the true value functions and not their estimates (equation 1), even if they must be estimated from data in practice.

---

> ### Author Response · Authors · 2023-11-16
>
> We thank the reviewer for all the valuable comments. Please refer to the point-to-point responses below.
>
> ##  W1: About the form of the upper confidence bound.
>
> Thank you very much for pointing out this inaccuracy. It is true that in linear MDPs where the transition kernel and reward function are assumed to be linear w.r.t. the state-action representation, the upper confidence bound has a specific form that does not correspond to Eq. (3) in our paper. We have corrected this part in our revised version (Section 3.1).
>
>
>
> ##  W2: About how the bootstrapped value ensemble is constructed and why we chose this form of uncertainty estimation.
>
> Specifically, we maintain $E$ bootstrapped $Q\_c$ networks, i.e., $\\{\widehat{Q}^i\_c \\}\_{i=1}^E$, which share the same architecture and only differ by the initial weights. From the Bayesian perspective, this ensemble forms an estimation of the posterior distribution of $Q\_c$, with the standard deviation quantifying the epistemic uncertainty induced by a lack of sufficient data. Despite its simplicity, this technique is widely adopted and is proved to be extremely effective in many previous works. We have added the above description into our revised version (Section 3.1).
>
>
>
> ##  W3: More space in the main paper for the theory behind ALM.
>
> Thanks for the suggestion. We have re-stated the monotonic energy decrease as a proposition and added a proof sketch of it in the main paper to provide an overall insight of the theory (Section 3.2).
>
>
>
> ##  Q: "How many Q networks did you use in the ensemble and how did you choose this hyperparameter?"
>
> As described in Appendix D.3, we set the ensemble size to 4 for MuJoCo tasks and 6 for Safety-Gym tasks. In theory, increasing the number of $Q\_c$ networks will improve the reliability of uncertainty quantification, but at the cost of increasing the computational burden. We choose the two hyperparameters because they strike a good balance between computational complexity and performance (we are running an additional ablation study for the ensemble size, and we will update our revision once we have completed experiments). However, note that we did not fine tune this hyperparameter for each task, so it is likely that a better choice of the ensemble size might exist for many tasks.

---

> > ### Author Response · Authors · 2023-11-21
> >
> > We have added an additional ablation study for the ensemble size $E$ in Appendix D.2 in our latest revision, and the corresponding analysis is in Section 4.2, highlighted in blue. Let us know if there are still any questions.

---

### Author Response · Authors · 2023-11-16

We appreciate all the valuable comments from the reviewers. We have revised our manuscript according to the suggestions and uploaded a modified version on this openreview webpage. Please refer to the revised part which is highlighted in blue.

---

### Meta-Review · Area_Chair_Z5G5 · 2023-12-12

**Metareview:**

In this paper, the authors proposed an algorithm for off-policy primal-dual safe policy learning. Specifically, the authors introduced UCB to overestimated Q-function, and thus improved conservatively. Meanwhile, the authors also exploit augmented Langrangian method to stabilize the algorithm performance. The algorithm is empirically evaluated on benchmarks.

**Justification For Why Not Higher Score:**

There are several issues in current version:

1, The theoretical analysis is hand-wavying. The uncrtainty quantification part is not rigorous.

2, The notation is quite confusing:
 - i), there is no \pi dependency in Q^UCB in Eq 4a.

 - ii), In Eq 4b, the argmin over \lambda is trivially 0. Why it still forms as a minmax problem?

3, The distribution shift issue in semi-batch setting has not been discussed.

**Justification For Why Not Lower Score:**

Most of the reviewers provide positive feedbacks and the empirical study shows the proposed algorithm is promising.

---

### Decision · Program_Chairs · 2024-01-16

Accept (poster)